# Glycolytic flux-signaling controls mouse embryo mesoderm development

**Hidenobu Miyazawa[1†], Marteinn T Snaebjornsson[1†‡], Nicole Prior[1§], Eleni Kafkia[2#], Henrik M Hammarén[2], Nobuko Tsuchida-Straeten[1¶], Kiran R Patil[2**], Martin Beck[2††], Alexander Aulehla[1*]**

[1]Developmental Biology Unit, European Molecular Biology Laboratory, Heidelberg, Germany; [2]Structural and Computational Biology Unit, European Molecular Biology Laboratory, Heidelberg, Germany

**\*For correspondence:**
aulehla@embl.de

†These authors contributed equally to this work

**Present address:** ‡German Cancer Research Center, Heidelberg, Germany; §School of Biological Sciences, University of Southampton, Southampton, United Kingdom; #Novo Nordisk Foundation Center for Stem Cell Biology, University of Copenhagen, Copenhagen, Denmark; ¶University Hospital Heidelberg, Heidelberg, Germany; **The Medical Research Council (MRC) Toxicology Unit,University of Cambridge, Cambridge, United Kingdom; ††Molecular Sociology, Max Planck Institute of Biophysics, Frankfurt, Germany

**Competing interest:** The authors declare that no competing interests exist.

**Abstract** How cellular metabolic state impacts cellular programs is a fundamental, unresolved question. Here, we investigated how glycolytic flux impacts embryonic development, using presomitic mesoderm (PSM) patterning as the experimental model. First, we identified fructose 1,6-bisphosphate (FBP) as an in vivo sentinel metabolite that mirrors glycolytic flux within PSM cells of post-implantation mouse embryos. We found that medium-supplementation with FBP, but not with other glycolytic metabolites, such as fructose 6-phosphate and 3-phosphoglycerate, impaired mesoderm segmentation. To genetically manipulate glycolytic flux and FBP levels, we generated a mouse model enabling the conditional overexpression of dominant active, cytoplasmic PFKFB3 (cytoPFKFB3). Overexpression of cytoPFKFB3 indeed led to increased glycolytic flux/FBP levels and caused an impairment of mesoderm segmentation, paralleled by the downregulation of Wnt-signaling, reminiscent of the effects seen upon FBP-supplementation. To probe for mechanisms underlying glycolytic flux-signaling, we performed subcellular proteome analysis and revealed that cytoPFKFB3 overexpression altered subcellular localization of certain proteins, including glycolytic enzymes, in PSM cells. Specifically, we revealed that FBP supplementation caused depletion of Pfkl and Aldoa from the nuclear-soluble fraction. Combined, we propose that FBP functions as a flux-signaling metabolite connecting glycolysis and PSM patterning, potentially through modulating subcellular protein localization.

## Editor's evaluation

How can generic changes in metabolism program specific changes in signaling and cell fate? To date it has been difficult to distill plausible models for specificity in this arena, but now Miyazawa et al. demonstrate a link between glycolytic flux, mesoderm segmentation and Wnt signaling. Enhanced flux translated into failures in segmentation and suppression of Wnt target gene expression, as well as inducing alterations to subcellular localization of glycolytic enzymes, suggesting pivotal links between glycolytic flux, signaling and lineage specification. Through careful work on presomitic mesoderm, the authors work suggests important new links between metabolism and differentiation.

## Introduction

Living systems have the critical ability to sense environmental cues, and to integrate this information with cellular functions by modulating their metabolic activity (*Efeyan et al., 2015*; *Kaelin and Ratcliffe, 2008*). The changes in metabolic activity, in turn, are sensed by multiple mechanisms to ensure that metabolic state matches cellular demands. Such mechanisms, referred to as metabolite sensing and signaling, generally consist of 'sentinel metabolites' and 'sensor molecules' (*Litsios et al.,*

2018; *Wang and Lei, 2018*). Sentinel metabolites mirror nutrient availability or cellular metabolic state by their levels. These metabolites can, in addition, potentially induce cellular responses, if their levels are linked to the activity of sensor molecules, such as proteins and RNAs. Well known examples of metabolite sensing and signaling include the mechanistic target of rapamycin (mTOR), which responds to altered levels of amino acids and couples nutritional availability with cell growth (*Saxton and Sabatini, 2017*), or AMP-activated protein kinase (AMPK), which senses adenosine monophosphate (AMP) levels and ensures that cellular bioenergetic demand matches cellular energetic state (*González et al., 2020*).

Importantly, the role of metabolite signaling is not limited to detecting nutrient availability to match metabolic activity and cellular demands. Recent work has highlighted the emerging link between central carbon metabolism and other cellular programs, such as gene regulation. For instance, by controlling the abundance of rate-limiting substrates used for post-translational modificiations, such as acetyl-CoA, metabolic activity can directly impact gene expression (*Campbell and Wellen, 2018*; *Reid et al., 2017*; *Miyazawa and Aulehla, 2018*). Glycolytic metabolites can also serve as signaling molecules that impact signal transduction directly. In yeast, for example, the glycolytic metabolite fructose 1,6-bisphosphate (FBP) has been shown to regulate the pro-proliferative RAS signaling cascade by interacting with the guanine nucleotide exchange factor Sos1 (*Peeters et al., 2017*). Notably, the connection between metabolic activity and other cellular programs can also occur at the level of metabolic enzymes with non-canonical, moonlighting functions (*Snaebjornsson and Schulze, 2018*; *Boukouris et al., 2016*; *Miyazawa and Aulehla, 2018*). In situations when moonlighting and canonical enzyme function are inter-dependent, a direct link between cellular metabolic state and moonlighting function is established. One such example is the glycolytic enzyme glyceraldehyde 3-phosphate dehydrogenase (Gapdh), which moonlights as an RNA-binding protein regulating translation when not engaged in its glycolytic function (*Chang et al., 2013*). While these studies highlight an intricate link between central carbon metabolism and other cellular functions, knowledge of metabolite signaling in more complex physiological settings, such as embryonic development, is still limited.

There are both classic (*Spratt, 1950*) as well as more recent findings (*Bulusu et al., 2017*; *Oginuma et al., 2017*; *Miyazawa et al., 2017*; *Bhattacharya et al., 2020*; *Djabrayan et al., 2019*; *Rodenfels et al., 2019*; *Chi et al., 2020*; *Miyazawa and Aulehla, 2018*) indicating that glucose metabolism and developmental programs are indeed linked. For instance, in mouse and chick embryos, the presomitic mesoderm (PSM) shows intrinsic differences in the expression levels of glycolytic enzymes, leading to the establishment of a glycolytic activity gradient along the anterior-posterior axis (*Bulusu et al., 2017*; *Oginuma et al., 2017*). The key question that remains largely unanswered is how a change in cellular metabolic activity is sensed and mechanistically linked to developmental programs.

To address this fundamental question, we focused on mouse embryos at the organogenesis stage following gastrulation, when glucose metabolism is rewired dynamically in time and space in response to extrinsic environmental cues and intrinsic developmental programs (*Miyazawa et al., 2017*; *Bulusu et al., 2017*; *Oginuma et al., 2017*). At this stage, the PSM is periodically segmented into somites, the precursors of vertebrae and skeletal muscles in vertebrates (*Hubaud and Pourquié, 2014*). PSM patterning and somite formation is controlled by the Wnt, FGF, and retinoic acid-signaling pathways, which show a graded activity along the anterior-posterior axis. In addition, PSM segmentation is linked to a molecular oscillator, the segmentation clock, comprised of several, interconnected signaling pathways (Notch, Wnt, Fgf) that show rhythmic activation cycles in PSM cells, with a period matching the rate of somite formation, for example ~2 hr in mouse embryos (*Aulehla et al., 2008*; *Yoshioka-Kobayashi et al., 2020*; *Soroldoni et al., 2014*; *Matsuda et al., 2020*; *Diaz-Cuadros et al., 2020*; *Chu et al., 2019*; *Sonnen et al., 2018*). The interplay between graded and oscillatory signaling dynamics within the PSM controls somite formation in time and space. Previously, a link between glycolytic activity and graded signaling activities has been found (*Bulusu et al., 2017*; *Oginuma et al., 2017*; *Oginuma et al., 2020*). In particular, evidence was found that glycolysis is part of a feedback loop linking (graded) FGF- and Wnt-signaling pathway activities (*Oginuma et al., 2017*; *Oginuma et al., 2020*). Although these studies revealed a link between glycolysis and morphogen signaling during PSM patterning, it remains unclear how a change in glycolytic activity is sensed and mechanistically linked to signaling.

In this study, our goal was therefore to first determine in vivo sentinel metabolites during mouse embryo PSM development. We then combined genetic, metabolomic and proteomic approaches

to investigate how altered glycolytic flux and metabolite levels impact developmental signaling and patterning processes.

## Results

### Steady state levels of FBP mirror glycolytic flux within PSM cells

In order to identify sentinel metabolites whose levels reflect glycolytic-flux within PSM cells, we quantified steady state metabolite levels in PSM samples cultured in various concentrations of glucose. We first verified that higher glucose concentrations led to higher glycolytic flux in PSM cells (*Figure 1A*). Throughout this study, we used quantification of secreted lactate as a proxy for glycolytic flux due to the inability to directly measure flux in embryonic tissues. We also analyzed somite formation and PSM patterning at different glucose concentrations using real-time imaging of the segmentation clock as a dynamic readout (*Figure 1—figure supplement 1*). PSM patterning proceeded normally, at least qualitatively, at glucose concentrations from 0.5 mM to 12.5 mM, with ongoing periodic morphological segmentation, axis elongation, and oscillatory clock activity throughout the PSM. Below or above this glucose range, morphological changes such as defects in PSM segmentation and axis elongation started to appear.

We hence focused on a glucose range between 0.5 and 10 mM to analyze steady state levels of metabolites in central carbon metabolism by gas chromatography mass spectrometry (GC-MS). Amongst the 57 metabolites quantified, 14 metabolites showed significant linear correlation (p-value <0.01) with extracellular glucose levels (*Figure 1B*). Fructose 1,6-bisphosphate (FBP) showed the highest correlation with extracellular glucose and also fold-change response to glucose titration (*Figure 1A*, *Figure 1C*). These results identify several sentinel metabolites, notably FBP, which had been shown to serve as a sentinel metabolite from bacteria to eukaryotic cell lines (*Kochanowski et al., 2013*; *Zhang et al., 2017*; *Peeters et al., 2017*; *Tanner et al., 2018*), in mouse embryos.

### Altered mesoderm development caused specifically by FBP supplementation

To test for a potential functional role of those sentinel metabolites that we identified, we next performed medium-supplementing experiments with the goal of altering intracellular metabolite levels. To this end, we supplemented the control culture medium with high levels of either fructose 6-phosphate (F6P), FBP, or 3-phosphoglycerate (3 PG) and scored the effect at the level of morphological segment formation, elongation, and also oscillatory segmentation clock activity, using real-time imaging quantifications. Interestingly, FBP supplementation impaired mesoderm segmentation and elongation and disrupted segmentation clock activity in the posterior PSM (*Figure 2A*, *Figure 2—figure supplement 1A*, *Figure 2—figure supplement 1B*). Immunostaining of active caspase-3 in explants did not reveal a major difference in cell death between control and FBP-treated explants (*Figure 2—figure supplement 1C*).

In contrast to the effects seen with FBP, glycolytic metabolites upstream (*i.e.* F6P) or downstream (*i.e.* 3 PG, pyruvate) of FBP did not cause such effects (*Figure 2A*, *Figure 2—figure supplement 1A*, *Figure 2—figure supplement 1B*; the effect of pyruvate supplementation was described in *Bulusu et al., 2017*). We also tested the effect of FBP supplementation on gene expression, focusing on an FGF-target gene *Dusp4* (*Niwa et al., 2007*) and a Wnt-target gene *Msgn1* (*Wittler et al., 2007*). Supplementation of FBP, but not F6P, caused a downregulation of *Dusp4* and *Msgn1* mRNA expression in a dose-dependent manner (*Figure 2B*), accompanying reduction of mesoderm segmentation and elongation (*Figure 2C*, *Figure 2D*). Of note, at intermediate concentration (10 mM) of FBP supplementation, only the Wnt-taget gene *Msgn1* was downregulated, while the Fgf-target gene *Dusp4* showed expression comparable to control samples, indicating potential dose-specific effects of FBP.

To validate the effects seen upon exogenous addition of FBP, we investigated the uptake of FBP by stable isotope ($^{13}$C) tracing. We cultured PSM explants in medium supplemented with fully $^{13}$C-labelled FBP ($^{13}$C$_6$-FBP) and analyzed $^{13}$C-labelling of intracellular metabolites by liquid chromatography mass spectrometry (LC-MS). Following three hours of incubation with $^{13}$C$_6$-FBP, $^{13}$C-labeling was detected in glycolytic intermediates downstream of FBP (*Figure 2—figure supplement 1D*), confirming the uptake of labeled carbons by the explants. Since we also detected that a small fraction

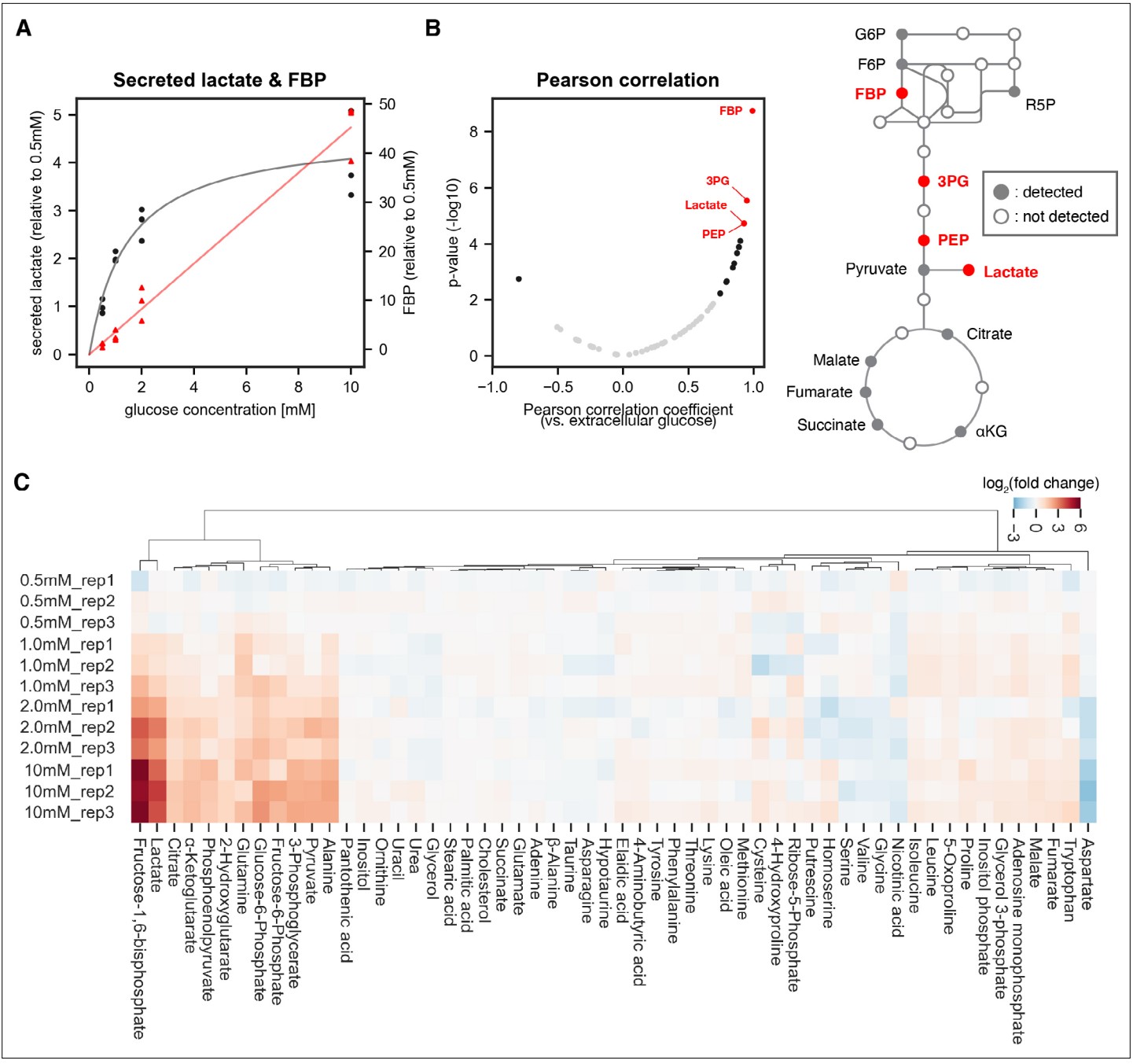

**Figure 1.** Identifying sentinel metabolites that mirror glycolytic flux. The amount of secreted lactate and intracellular metabolites within PSM explants were measured by gas chromatography mass spectrometry (GC-MS; n=3 biological replicates for each condition). The explants were cultured for 3 hr ex vivo in 0.5 mM, 1.0 mM, 2.0 mM, or 10 mM glucose. (**A**) The relative amount of secreted lactate (shown in black circles) and intracellular fructose 1,6-bisphosphate (FBP; shown in red triangles) under various glucose conditions. The gray and red lines show the Michaelis-Menten fit (Vmax = 4.7 arbitrary unit, Km = 1.5 mM) for secreted lactate and the linear regression line for intracellular FBP, respectively. (**B**) Pearson correlation analysis between intracellular metabolite levels and extracellular glucose levels. Metabolites showing significant correlation (p-value < 0.01) are shown in black. Those with a |Pearson correlation coefficient|>0.9 are highlighted in red. Abbreviations: G6P, glucose 6-phosphate; F6P, fructose 6-phosphate; R5P, ribose 5-phosphate; FBP, fructose 1,6-bisphosphate; 3 PG, 3-phosphoglycerate; PEP, phosphoenol pyruvate; αKG, α-ketoglutarate. (**C**) Hierarchical clustering heatmap of metabolites detected in the PSM explants. Fold changes were calculated using 0.5 mM glucose condition as the reference. Hierarchical clustering was performed using Ward's method with Euclidean distance.

The online version of this article includes the following figure supplement(s) for figure 1:

**Figure supplement 1.** Effects of glucose titration on PSM patterning.

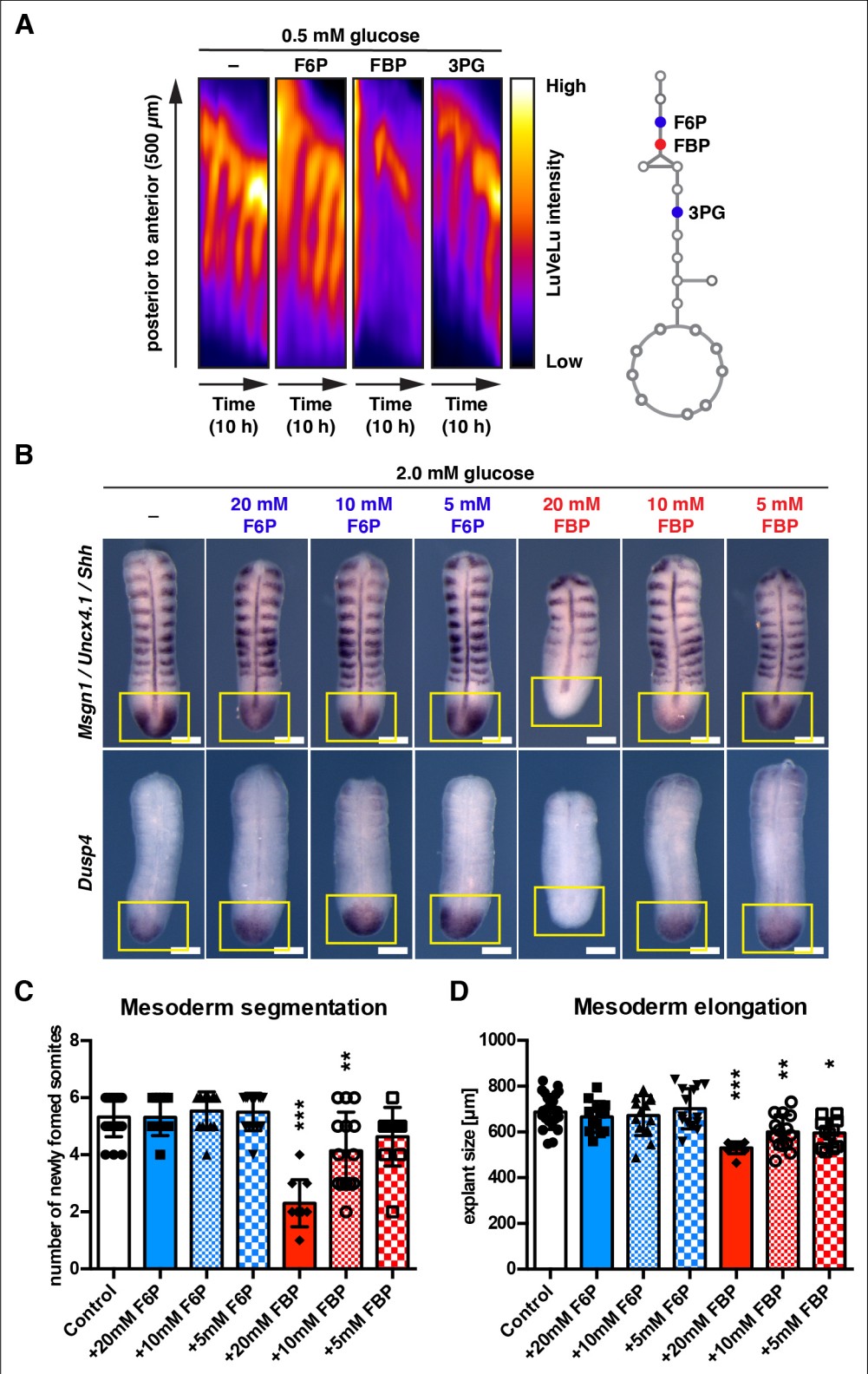

**Figure 2.** FBP supplementation impacts mesoderm segmentation and elongation in a dose-dependent manner. (**A**) Kymographs showing dynamics of the Notch signaling activity reporter LuVeLu in PSM explants treated with 20 mM of the indicated metabolite. (**B**) Whole mount in situ hybridization analysis for the FGF (*i.e. Dusp4*) and the Wnt target (*i.e. Msgn1*) gene expression in the PSM. PSM explants were incubated for 12 hr in the presence of F6P

*Figure 2 continued*

or FBP. Expression domains of *Dusp4* and *Msgn1* are indicated by yellow squares. *Shh* and *Uncx4.1* were used as markers for the neural tissue and posterior somite boundary, respectively. Scale bar, 200 µm. (**C, D**) The number of newly formed somites (**C**) and the length of PSM explants (**D**) after 12 hr ex vivo culture (one-way ANOVA with Tukey's post-hoc test, *p-value <0.05, **p-value <0.01, ***p-value <0.001 versus control).

The online version of this article includes the following figure supplement(s) for figure 2:

**Figure supplement 1.** Effects of medium-supplementation of glycolytic intermediates on mesoderm elongation and segmentation.

**Figure supplement 2.** Modulation of Wnt-target gene expression upon glucose titration within PSM cells.

of $^{13}C_6$-FBP broke down to $^{13}C_6$-fructose monophosphate (F6P and/or fructose 1-phosphate (F1P)) in the culture medium during incubation (data not shown), we performed additional control experiments by culturing PSM explants in F1P-supplemented medium. Similar to F6P, supplementation of F1P did not cause any detectable phenotype at the level of segmentation clock activity or elongation (*Figure 2—figure supplement 1E*).

As a related finding, we observed that upon glucose titration, the expression of Wnt-signaling target genes in PSM explants is anti-correlated with glucose availabilty/glycolytic activity: while lowering glucose concentration (from 5.0 mM to 0.5 mM) correlated with an upregulation of several Wnt target genes, such as *Axin2*, *Ccnd1*, and *Myc*, the opposite effect was found when glucose concentration was increased (from 5.0 mM to 25 mM) (*Figure 2—figure supplement 2*).

Combined, our findings hence suggest that FBP, but not other glycolytic intermediates such as F6P, F1P, or 3 PG, is a flux-sentinel and signaling metabolite, as it impacts mesoderm development and gene expression in a dose-dependent manner.

## Generating a conditional *cytoPFKFB3* transgenic mouse line as a genetic tool to increase glycolytic flux

Our findings thus far show that intracellular FBP levels respond dynamically to an alteration in glycolytic flux (*Figure 1*), and importantly, that FBP, but not its precursor metabolite F6P, impacts PSM development in a dose-dependent manner (*Figure 2*). Based on these observations, we next sought a way to manipulate glycolytic flux at the level of the phosphofructokinase (Pfk) reaction and importantly, in a genetic manner (*Figure 3A*). Pfk converts F6P into FBP, the first committed step in glycolysis, and plays a critical role in regulating glycolytic flux (*Tanner et al., 2018*; *Mor et al., 2011*). We generated transgenic mice enabling conditional overexpression of a mutant PFKFB3 *i.e.* PFKFB3(K472A/K473A) (*Yalcin et al., 2009*). PFKFB3 generates fructose 2,6-bisphosphate (F2,6BP), a potent allosteric activator of Pfk (*Figure 3A*). A previous study showed that PFKFB3(K472A/K473A) localises exclusively to the cytoplasm, and that this cytoplasmically-localized PFKFB3 (hereafter termed as cytoPFKFB3) activates glycolysis (*Yalcin et al., 2009*). Indeed, in PSM explants from transgenic embryos with ubiquitous overexpression of cytoPFKFB3, we found increased glycolysis based on the analysis of lactate secretion (*Figure 3B*). In addition, we found that in *cytoPFKFB3* embryos, lactate secretion changed in a glucose-dose dependent manner (*Figure 3B*). Next we investigated steady state metabolite levels in control and transgenic PSM explants cultured in 10 mM glucose condition. Among the 57 metabolites quantified by GC-MS, FBP and lactate were significantly increased in transgenic PSM explants, while aspartate, glucose 6-phosphate, and glutamate were significantly decreased (*Figure 3C*, *Figure 3—figure supplement 1A*). These findings mirrored the results in wild-type PSM explants upon glucose titration (*Figure 1*).

It is notable that cytoPFKFB3 overexpression enables glycolytic flux to reach a level that is not achievable in control embryos (*Figure 3B*). Consistently, we found that cytoPFKFB3 overexpression lifted the upper limit of FBP levels in PSM cells (*Figure 3E*, *Figure 3—figure supplement 1B*, *Figure 3—figure supplement 1C*). In control explants, FBP levels did not increase further when glucose concentration was increased from 10 mM to 25 mM. It was also the case when control explants were cultured in 20 mM of F6P (*Figure 3E*). These results indicate that the Pfk reaction carries a (rate-)limiting role for glycolytic flux and FBP levels, and that cytoPFKFB3 overexpression hinders the flux-regulation function of Pfk. As a possible indicator of dysregulated flux at the level of Pfk reaction, we observed that the ratio between FBP and glucose mono-phosphate (G6P/F6P) was increased in cytoPFKFB3

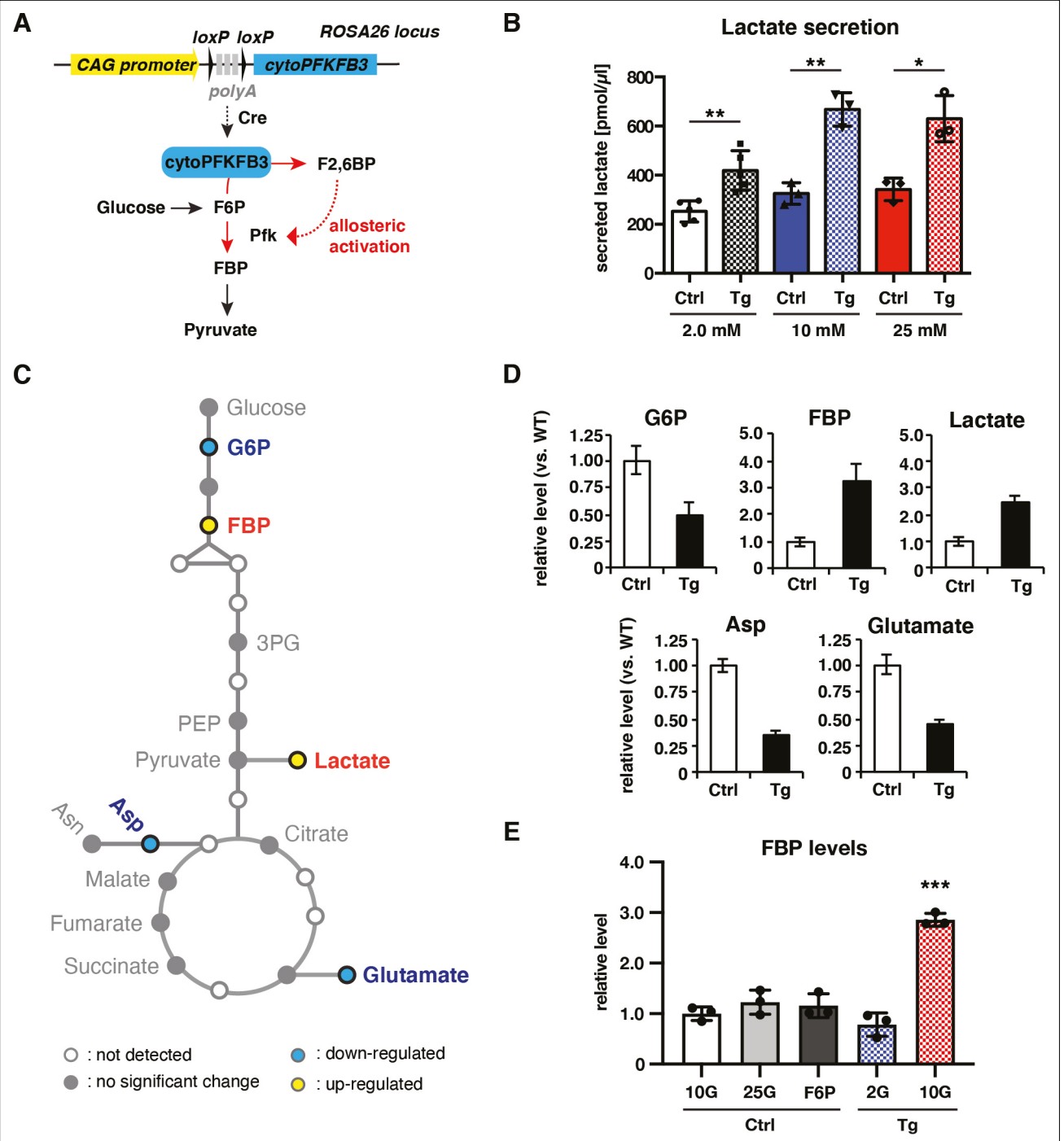

**Figure 3.** cytoPFKFB3 overexpression causes an increase in glycolytic flux and FBP levels within PSM cells. (**A**) Conditional *cytoPFKFB3* transgenic mice were generated to activate glycolysis through allosteric activation of Pfk. (**B**) Quantification of secreted lactate in control and *cytoPFKFB3* transgenic PSM explants cultured for 12 hr under varying concentrations of glucose (unpaired Welch's *t*-test, *p-value <0.05, **p-value <0.01). (**C, D**) Measurement of steady state metabolite levels by GC-MS (n=4 biological replicates for each condition) in control (Ctrl) and *cytoPFKFB3* (Tg; crossed to *Hprt^Cre* line) explants cultured for 3 hr in medium containing 10 mM glucose. SAM (Significance Analysis for Microarrays) analysis was performed using a significance threshold $\delta = 0.9$, which corresponds to a false discovery rate (FDR)=0.012. G6P, glucose 6-phosphate. FBP, fructose 1,6-bisphosphate. 3 PG, 3-phosphoglycerate. PEP, phosphoenol pyruvate. Asp, aspartate. Asn, asparagine. (**E**) Targeted metabolomics analysis by liquid chromatography-mass spectrometry (LC-MS). Relative FBP levels were determined in control and *cytoPFKFB3* explants cultured for 3 hr in various culture conditions (2 G: 2.0 mM glucose, 10 G: 10 mM glucose, 25 G: 25 mM glucose, F6P: 2.0 mM glucose plus 20 mM F6P; n=3 biological replicates for each culture condition). Unpaired Welch's *t*-test (***p-value <0.001 vs. Ctrl-10G).

*Figure 3 continued on next page*

*Figure 3 continued*

The online version of this article includes the following figure supplement(s) for figure 3:

**Figure supplement 1.** Steady state measurements of metabolites within *cytoPFKFB3* and control PSM explants by GC-MS.

embryos compared to control even when FBP levels were comparable between them (*Figure 3—figure supplement 1D*).

We hence conclude that the overexpression of cytoPFKFB3 leads to activation of glycolysis at the level of Pfk in a glucose-dose dependent manner. More generally, the *cytoPFKFB3* transgenic mouse line represents a potentially powerful new genetic model to study the role of glycolysis.

## Functional consequence of cytoPFKFB3 overexpression on PSM development

We then investigated the functional consequences of cytoPFKFB3 overexpression on mesoderm development. Constitutive overexpression of cytoPFKFB3 from fertilization caused embryonic lethality, as no transgenic pups were recovered (n=30 pups, N=6 litters). We have not yet investigated the precise timepoint and cause of lethality. At embryonic day 10.5 (E10.5), cytoPFKFB3 transgenic embryos were morphologically indistinguishable from their littermates, but had slightly fewer somites (*Figure 4A*; control: 38±1.5 somites, transgenic: 35±3.9 somites).

To analyze the impact of cytoPFKFB3 overexpression on mesoderm development in a more dynamic and quantitative manner, we analyzed mesoderm segmentation, elongation, and oscillatory clock activity in *cytoPFKFB3* and control explants cultured at various glucose concentrations. Consistent with our previous findings (*Figure 1—figure supplement 1*), control explants proceeded segmentation and PSM patterning in a qualitatively comparable manner, even when cultured at higher glucose concentrations (*Figure 4B*). In contrast, we found that somite formation was impaired in explants from *cytoPFKFB3* embryos in a glucose-dose dependent manner (*Figure 4B*). Overall growth during this 12 hr incubation seemed comparable or even increased in *cytoPFKFB3* transgenic explants, based on the size of explants after culture (*Figure 4C*). We also tested whether a mesoderm-specific cytoPFKFB3 overexpression has a similar effect on somite formation. Indeed, mesoderm specific cytoPFKFB3 overexpression, using Cre-expression driven by the promoter of the pan-mesoderm marker *Brachyury* (i.e. *T*-promoter-driven Cre *Perantoni et al., 2005*), showed similar reduction in segment formation, compared to control explants (*Figure 4E*, *Figure 4F*). The real-time imaging quantification of segmentation clock activity revealed that in *cytoPFKFB3* explants cultured at 10 mM glucose, clock oscillations ceased after few cycles, in contrast to control samples (*Figure 4G*; *Figure 4—video 1*).

Molecularly, we found that the expression of the Wnt signaling target gene *Msgn1* was downregulated in *cytoPFKFB3* explants, again in a glucose-concentration dependent manner (*Figure 5A*). Of great interest, about 30% of *cytoPFKFB3* explants showed reduced expression of *Msgn1* even under 2.0 mM glucose condition where their FBP levels are within the range of wild-type explants (*Figure 3—figure supplement 1B*). In contrast, we did not find an obvious change in the expression of *Dusp4*, an Fgf signaling target, which was maintained even at 25 mM glucose (*Figure 5B*).

To elaborate these findings, we next performed a transcriptome analysis of control and *cytoPFKFB3* explants cultured in 10 mM glucose for three hours. We identified 568 genes as differentially expressed genes (DEGs; adjusted p-value <0.01; *Supplementary file 1*): 210 genes were upregulated in *cytoPFKFB3* explants, while 358 genes were downregulated (*Figure 5C*). Genes associated with transcription, anterior-posterior patterning, and the Wnt signaling pathway were enriched among the downregulated DEGs (*Supplementary file 2*), and those DEGs included many Wnt-target genes (*Figure 5C*). No gene ontology (GO) term was enriched among the upregulated DEGs.

To examine whether FBP addition would mirror effects on gene expression and in particular Wnt-signaling target genes, we then performed a transcriptome analysis of explants cultured with FBP (*Figure 5D*). While FBP supplementation caused upregulation of cell cycle- or metabolism-related genes, it led to downregulation of genes associated with transcription and anterior-posterior patterning (*Supplementary file 3*; *Supplementary file 4*). Of great importance, these downregulated DEGs included many Wnt-targets (*Figure 5D*), most of which were also downregulated in *cytoPFKFB3* explants. F6P-treated explants did not show such feature (*Figure 5E*, *Supplementary file 5*;

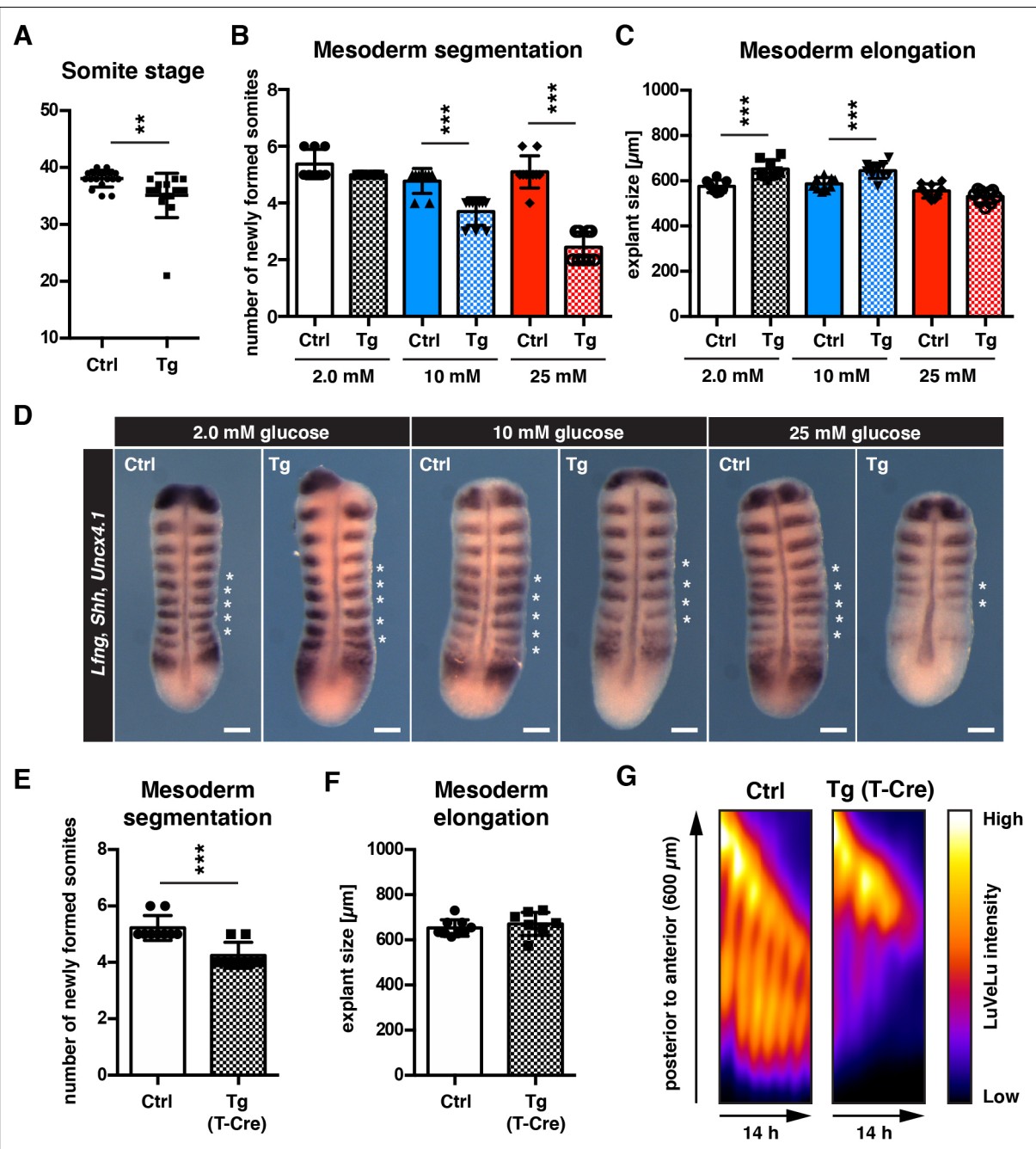

**Figure 4.** cytoPFKFB3 overexpression impacts mesoderm development in a glucose-concentration dependent manner. (**A**) Total number of somites in E10.5 embryos (mean ±s.d; unpaired Welch's *t*-test; **p-value <0.01). Ctrl, control embryos; Tg, *cytoPFKFB3* embryos (crossed to *Hprt^Cre* line). (**B,C**) Number of formed somites and quantification of PSM explant length after 12 hr in vitro culture. (**D**) Whole mount mRNA in situ hybridization analysis for *Lfng, Shh*, and *Uncx4.1* in PSM explants after 12 hr in vitro culture at varying glucose concentrations. Asterisks denote somites that formed during the in vitro culture. Scale bar, 100 µm. (**E,F**) Effect of mesoderm-specific overexpression of cytoPFKFB3 on PSM segmentation and elongation (12 hr incubation). The PSM explants were cultured in medium containing 10 mM glucose. Bar graphs show the number of newly formed somites during the culture (**E**), and the length of explants after the culture (**F**; mean ±s.d; unpaired Welch's *t*-test; ***p-value <0.001). (**G**) Real-time quantification of segmentation clock activity using Notch signaling activity reporter LuVeLu in PSM explants, shown as kymographs. Note that oscillatory reporter activity ceased in cytoPFKFB3/T-Cre samples during the experiment, while control samples showed ongoing periodic activity.

The online version of this article includes the following video and figure supplement(s) for figure 4:

**Figure supplement 1.** Phenotype of *cytoPFKFB3* embryos in vivo is dependent on maternal glucose conditions.

**Figure 4—video 1.** Real-time imaging of control and cytoPFKFB3/T-Cre explants expressing the Notch signaling reporter LuVeLu.
https://elifesciences.org/articles/83299/figures#fig4video1

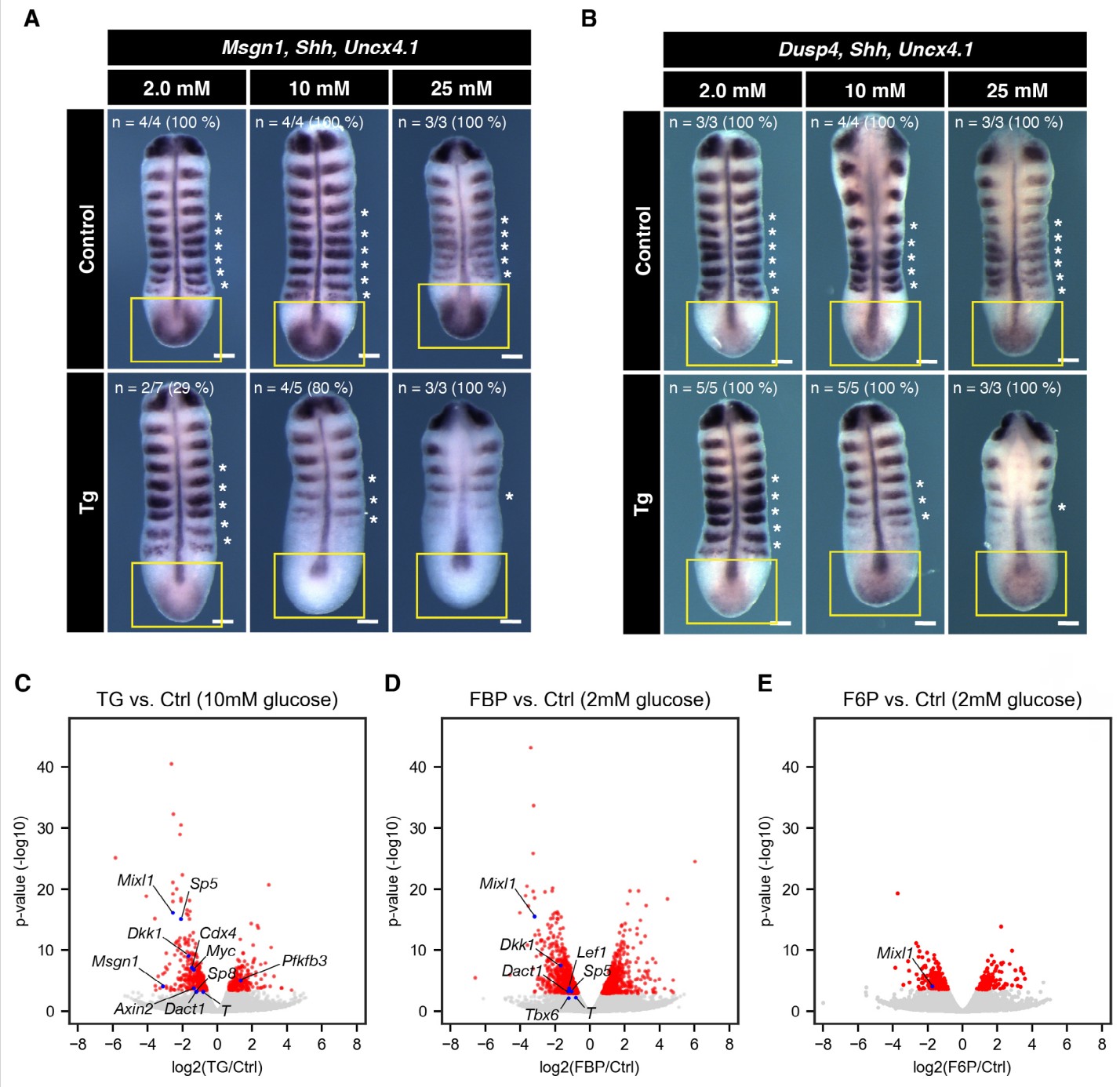

**Figure 5.** Effect of cytoPFKFB3 overexpression on Wnt and FGF target gene expression. (**A,B**) Whole mount mRNA in situ hybridization for *Msgn1* (Wnt-target gene) and *Dusp4* (FGF-target gene) in the PSM explants. Explants were cultured for 12 hr under various glucose conditions, as indicated. *Shh* and *Uncx4.1* were used as markers for neural tissue and posterior somite boundary, respectively. Expression domain of *Msgn1* and *Dusp4* is indicated by yellow rectangles. Note the glucose-dose dependent loss of *Msgn1* expression in *cytoPFKFB3* explants (Tg; crossed to *Hprt*[Cre] line). In contrast, *Dusp4* expression appeared unaffected in *cytoPFKFB3* explants. Asterisks mark somites that formed during the culture. Scale bar, 100 μm. (**C–E**) Transcriptome analysis of PSM explants cultured for 3 hr in vitro (n=3 biological replicates for each culture condition). Gene expression profiles were compared between control and *cytoPFKFB3* explants (**C**), control and FBP (20 mM)-treated explants (**D**), or control and F6P (20 mM)-treated explants (**E**). Among differentially expressed genes (adjusted p-value <0.01; shown in red), Wnt-target genes and PFKFB3 were marked by blue.

*Supplementary file 6*). Therefore, these results indicate that an increase in glycolytic flux or FBP levels leads to suppression of Wnt signaling activity.

Combined, these results show that cytoPFKFB3 overexpression results in reduced segment formation, arrest of the segmentation clock oscillations and downregulation of Wnt signaling, in a glucose-dose dependent manner. As glucose concentration impacts, in turn, glycolytic flux (*Figure 3B*), these findings suggest that these phenotypes are flux-dependent and are not a mere result of cytoPFKFB3 overexpression. In addition, we found that exogenous FBP-supplementation likewise causes a dose dependent effect on clock oscillations and downregulation of Wnt-signaling target gene expression (*Figure 2*, *Figure 5D*), implicating FBP as a mediator of flux-sensitive effects on development and signaling.

## In vivo phenotype of *cytoPFKFB3* embryos is sensitive to maternal environment

As noted above, *cytoPFKFB3* embryos were morphologically indistinguishable from control littermates when dissected at E10.5. This contrasts with the PSM phenotype we found when *cytoPFKFB3* PSM explants were cultured in vitro (*Figure 4*, *Figure 5*). As this phenotype is glucose dose dependent, we reasoned that the absence of an obvious in vivo phenotype at E10.5 could reflect low in vivo glucose concentrations, which have been reported to be lower than in maternal circulation (*Renfree et al., 1975*). To test this possibility, we performed whole embryo roller-culture (WEC) experiments with cytoPFKFB3 embryos at E8.5, exposing them to ~5 mM glucose (50% rat serum / DMEM with 1.0 g/L glucose). Indeed, while all control embryos completed cranial neural tube closure (NTC) (n=12/12) after 24 hr WEC, about 40% of the transgenic embryos (n=7/18) failed to complete this process, showing a developmental delay as well (*Figure 4—figure supplement 1A–C*).

In order to further test the hypothesis in vivo we next used the maternal diabetes mouse model Akita (*Wang et al., 1999*; *Yoshioka et al., 1997*). Akita mice carry a point mutation in the *Ins2* gene, which leads to a diabetic phenotype including hyperglycemia. Akita heterozygous females indeed showed elevated blood glucose levels (i.e. ~450 mg/dl) compared to control (i.e. ~150 mg/dl). On the maternal diabetic background, 50% of cytoPFKFB3 embryos (n=5 out of 10 embryos) showed neural tube defects (NTDs) with developmental delay in vivo, while less than 10% of control embryos (n=1 out of 13 embryos) showed NTDs (*Figure 4—figure supplement 1D*, *Figure 4—figure supplement 1E*). In addition, they had fewer somites than control embryos. This provides in vivo evidence for a glycolytic flux-dependent impact on embryonic development in *cytoPFKFB3* embryos.

## Perturbation of glycolytic-flux and FBP levels alters subcellular localization of glycolytic enzymes

Our data thus far suggest that altered glycolysis, caused by either nutritional or genetic means, impairs PSM development, possibly mediated via the sentinel metabolite FBP. To probe for potential underlying mechanisms, we turned to the role of glycolytic enzymes. Interestingly, we had found that several glycolytic enzymes are localized in the nucleus in PSM cells, based on cell-fractionation analysis (*Figure 6—figure supplement 1C*, *Figure 6—figure supplement 1D*). It had been proposed previously that the subcellular localization of glycolytic enzymes can change dynamically in response to altered glycolytic flux (*Kwon et al., 2010*; *Hu et al., 2016*; *Zhang et al., 2017*). We therefore aimed to systematically investigate the changes in subcellular protein localization in response to altered metabolic state in mouse embryos. To this end, we performed a proteome-wide cell-fractionation analysis in PSM explants cultured in various metabolic conditions.

Proteins were extracted from cytoplasmic, membrane, nuclear-soluble, chromatin-bound, and the remaining insoluble (labeled as 'cytoskeletal') fractions. We found that in samples cultured for three hours in FBP-supplemented medium (and to a lesser extend in F6P-supplemented medium), proteins part of the glycolytic pathway (12 combined glycolytic enzymes) were reduced in the cytoskeletal and, to a lesser extent, the nuclear soluble fraction, relative to samples cultured in control medium (*Figure 6A*, *Figure 6—figure supplement 1A*, *Figure 6—figure supplement 1B*). For several glycolytic enzymes detected in the nuclear soluble fraction, that is aldolase A (Aldoa), phosphofructokinase L (Pfkl), glyceraldehyde 3-phosphate dehydrogenase (Gapdh), and pyruvate kinase M (Pkm) (*Figure 6—figure supplement 1E*), we performed a targeted analysis using Western blotting

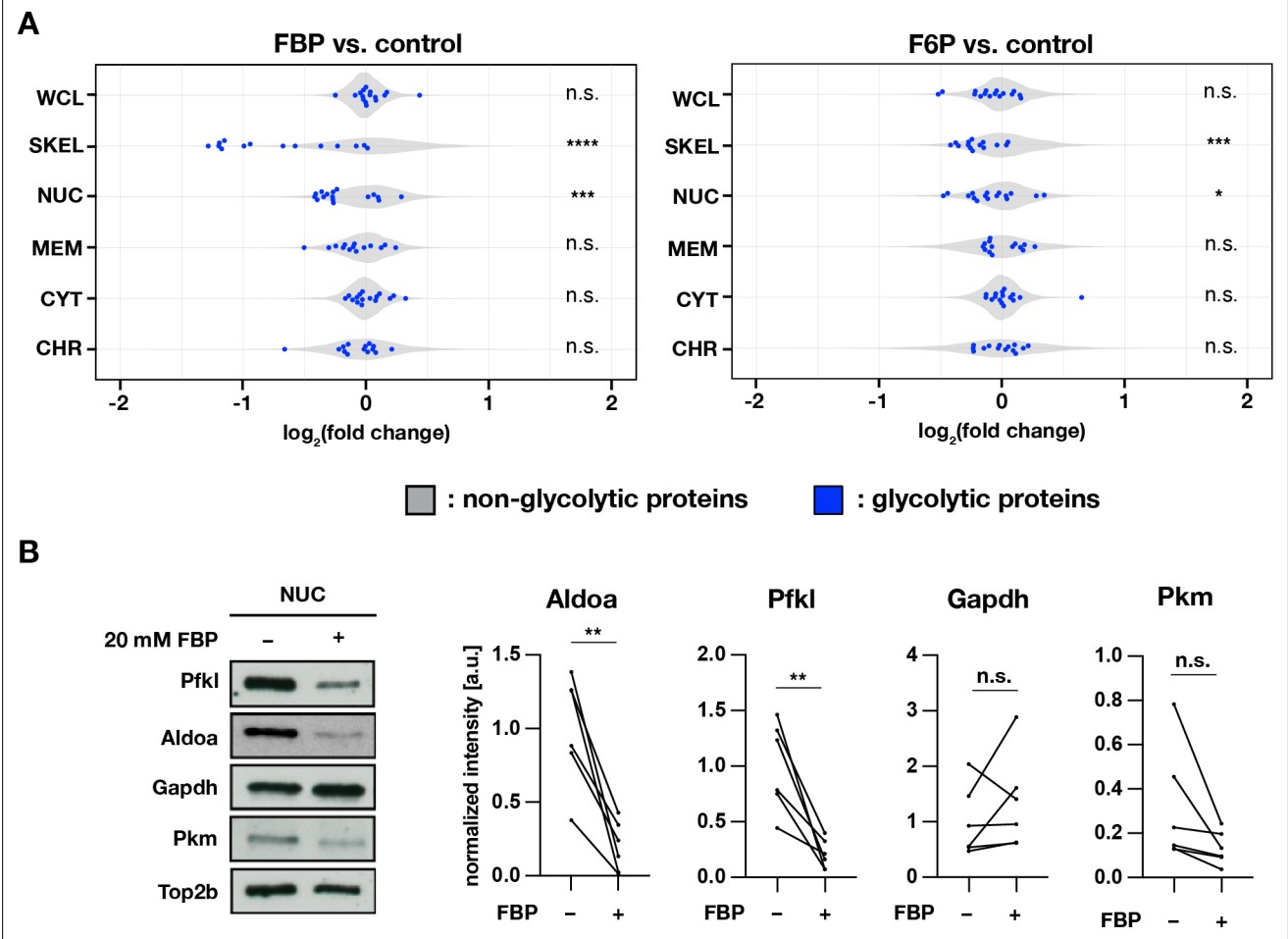

**Figure 6.** Subcellular localization of glycolytic enzymes are responsive to FBP treatment. (**A**) Effects of FBP treatment on subcellular localization of glycolytic enzymes. PSM explants were cultured for 3 hr in media containing 2.0 mM glucose and supplemented with 20 mM F6P or FBP. In addition to whole cell lysates (WCL), protein extracts were prepared from cytoplasmic (CYT), membrane (MEM), nuclear-soluble (NUC), chromatin-bound (CHR), and cytoskeletal (SKEL) fractions (n=3 biological replicates). Abundance ratios (log$_2$(F6P/FBP-treated/control)) of glycolytic enzymes (in blue) were compared to those of non-glycolytic proteins (the rest, in gray) for statistical analysis (unpaired two-sample Wilcoxon test, *p-value <0.05, ***p-value <0.001, ****p-value <0.0001, n.s., not significant). (**B**) Effects of FBP on the abundance of glycolytic enzymes in the nuclear soluble fraction. Subcellular protein fractionation was performed following 1 hr incubation of PSM explants in the media containing 0.5 mM glucose and supplemented with 20 mM FBP (n=6 biological replicates; paired *t*-test, **p-value <0.01, n.s., not significant).

The online version of this article includes the following source data and figure supplement(s) for figure 6:

**Source data 1.** Uncropped, unedited blots for *Figure 6B*.

**Figure supplement 1.** Proteome analysis of subcellular protein localization in the PSM.

**Figure supplement 1—source data 1.** Uncropped, unedited blots for *Figure 6—figure supplement 1E*.

(*Figure 6B*). Interestingly, we found that amongst those tested enzymes, Aldoa and Pfkl were significantly depleted from the nuclear soluble fraction upon incubation in FBP-supplemented medium.

We next asked whether subcellular localization of glycolytic enzymes is also altered upon cyto-PFKFB3 overexpression, which we showed leads to an increase in glycolytic flux and FBP levels (*Figure 3B–D*). We hence performed subcellular proteome analysis of both control and *cytoPFKFB3* transgenic PSM explants, cultured for 1 hr in 10 mM glucose-containing medium. Due to the limited material obtained from transgenic embryos, proteins from nuclear-soluble, chromatin-bound, and cytoskeletal fractions were collected as a single, nuclear-cytoskeletal fraction. We found that cyto-PFKFB3 overexpression altered the nuclear-cytoskeletal abundance of 12 proteins among 2813 detected proteins (adjusted p-value <0.05 and |log$_2$(fold change)|>0.5) (*Figure 7A–C*). One of these proteins showing a pronounced depletion in the nuclear-cytoskeletal fraction in transgenic explants

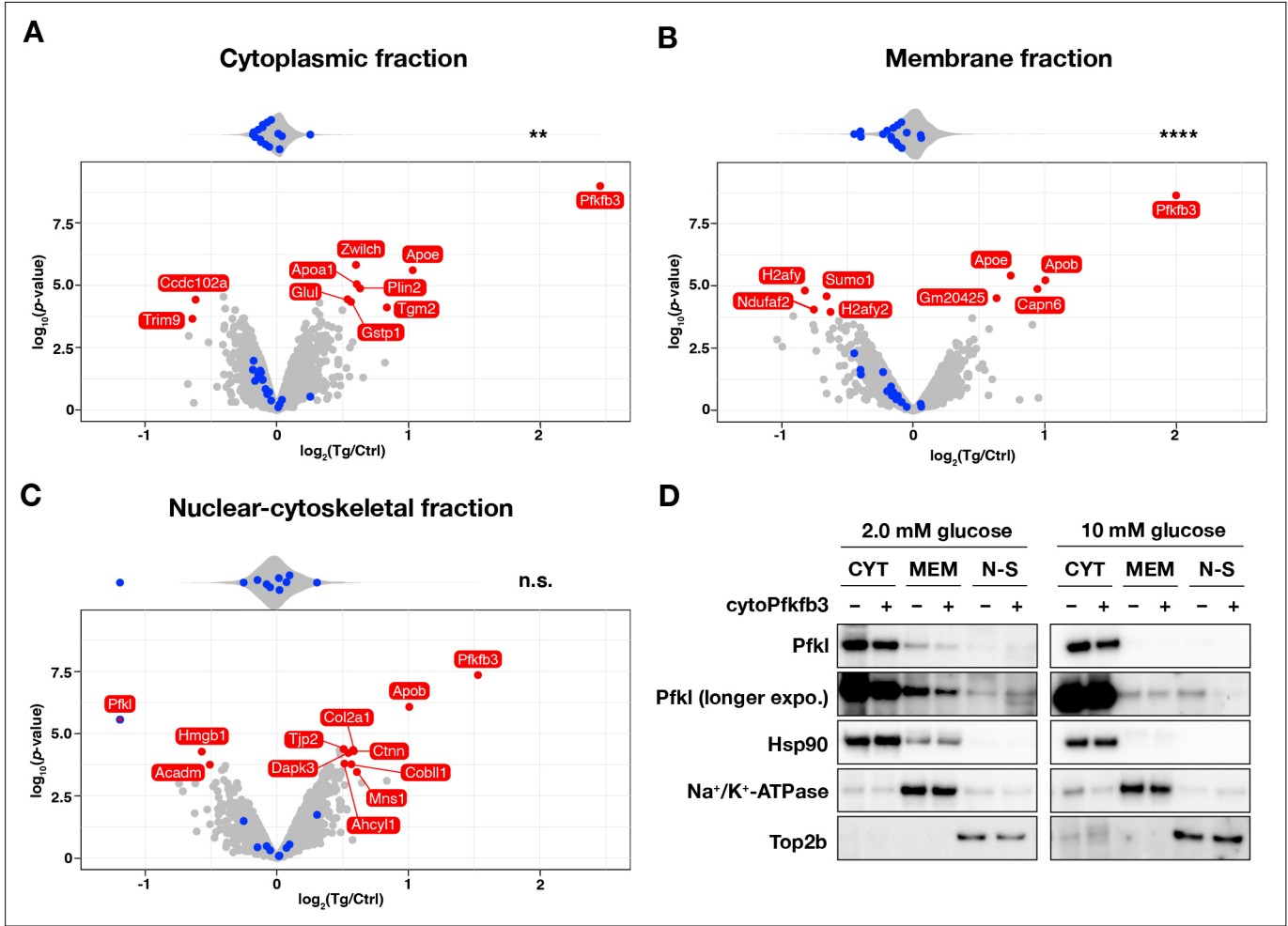

**Figure 7.** Subcellular localization of Pfkl responds to cytoPFKFB3 overexpression in a glucose-concentration dependent manner. (**A–C**) Effects of cytoPFKFB3 overexpression on subcellular protein localization assessed by mass spectrometry. Following 1 hr incubation of PSM explants in 10 mM glucose, protein extracts were prepared from cytoplasmic (**A**), membrane (**B**), and nuclear-cytoskeletal (**C**) fractions (n=3 biological replicates for each culture condition). Proteins whose abundance showed significant changes (adjusted p-value <0.05 and |log$_2$(fold change)|>0.5) are marked red in the volcano plots. Top violin plots show distribution of abundance changes of glycolytic (blue) and non-glycolytic (gray) proteins. Statistical comparison as in *Figure 6*. **p-value <0.01. Ctrl, control explants; Tg, *cytoPFKFB3* explants (crossed to the *Hprt$^{Cre}$* line). (**D**) Western-blot analysis of subcellular localization of Pfkl under different glucose conditions. Subcellular protein fractionation was performed following 1 hr incubation of PSM explants under 2.0 mM or 10 mM glucose (n=3 biological replicates for each culture condition). CYT, cytoplasmic fraction; MEM, membrane fraction; N-S, nuclear-cytoskeletal fraction.

The online version of this article includes the following source data for figure 7:

**Source data 1.** Uncropped, unedited blots for *Figure 7D*.

turned out to be the glycolytic enzyme Pfkl (*Figure 7C*). Using western blotting, we confirmed that Pfkl was depleted in the nuclear-cytoskeletal fraction in transgenic explants cultured at 10 mM glucose (*Figure 7D*). Importantly, under 2.0 mM glucose condition, nuclear-cytoskeletal Pfkl was not depleted in transgenic explants, suggesting that subcellular localization of Pfkl changes in a glucose-dose-dependent manner. In addition, we found that, in *cytoPFKFB3* explants, the overall abundance of glycolytic machinery was decreased in the cytoplasmic and membrane fraction (*Figure 7A–B*).

Combined, our results hence reveal that an alteration in glycolytic-flux/FBP levels, either by direct supplementation of metabolites or by genetic means using cytoPFKFB3 overexpression, changes the distribution of glycolytic enzymes in several subcellular compartments. While we have not been able to address the functional consequence of specific changes in subcellular localization, such as the nuclear depletion of Pfkl or Aldoa when glycolytic flux is increased, these results pave the way for

future investigations on the mechanistic underpinning of how metabolic state is linked to cellular signaling and functions.

## Discussion

### Identifying FBP as a sentinel metabolite for glycolytic flux in developing mouse embryos

In this work, we investigated how glycolytic flux impacts mouse embryo mesoderm development, seeking to decipher the underlying mechanisms. First, we aimed to identify sentinel metabolites whose concentrations mirror glycolytic flux in mouse embryos (*Kochanowski et al., 2013*; *Zhang et al., 2017*; *Cai et al., 2011*; *Peeters et al., 2017*). The identification of sentinel metabolites is critical, as steady state metabolite levels are generally poor indicators of metabolic pathway activities (*Jang et al., 2018*). By investigating how steady state metabolite levels respond to an alteration in glycolytic flux upon glucose titration, we identified aspartate, FBP, and lactate as potential sentinel glycolytic metabolites whose steady state levels were either positively (*i.e.* FBP and lactate) or negatively (*i.e.* aspartate) correlated with extracellular glucose levels (*Figure 1B*). Similar changes were observed upon glycolytic activation by cytoPFKFB3 overexpression (*Figure 3C*). Remarkably, we found that FBP levels exhibit a strong linear correlation with a wide range of glucose concentrations, showing a 45-fold increase from 0.5 mM to 10 mM glucose conditions (*Figure 1A*). In addition, FBP levels showed a linear correlation with lactate secretion in control explants, and such a correlation was maintained even in *cytoPFKFB3* explants (*Figure 3—figure supplement 1C*). Previous studies suggested that the reversible reactions between FBP and PEP allow coupling of FBP to lower glycolytic flux (*Kochanowski et al., 2013*), and importantly that feedforward activation of pyruvate kinase by FBP enables the cell to establish a linear correlation between FBP and glycolytic flux over a wide range of FBP concentrations (*Kotte et al., 2010*; *Kochanowski et al., 2013*). Such properties of lower glycolytic reactions may allow FBP to function as a generic sentinel metabolite for glycolytic flux in various biological contexts, from bacteria to mammalian cells (*Kochanowski et al., 2013*; *Zhang et al., 2017*; *Peeters et al., 2017*; *Tanner et al., 2018*). This study extends such a finding of FBP as a glycolytic sentinel metabolite to in vivo mammalian embryos.

### FBP as a flux signaling metabolite connecting glycolytic-flux and PSM development

Interestingly, in addition to being a sentinel for glycolytic flux, FBP has been shown to carry signaling functions, hence relaying flux information to downstream effectors, such as transcription factors and signaling molecules (*Kochanowski et al., 2013*; *Zhang et al., 2017*; *Peeters et al., 2017*). To test if such a flux-signaling function exists also in mouse embryos, we combined two complementary approaches, that is medium-supplementation of FBP (*Figure 2*) and, importantly, a genetic mouse model to increase glycolytic flux (*Figures 3–5*).

First, we revealed that high doses of FBP impaired mesoderm segmentation, disrupted the segmentation clock activity and led to downregulation of Wnt and Fgf target gene expression in the PSM (*Figure 2*). Using $^{13}$C-tracing experiments, we showed that exogenous FBP could be taken up by PSM cells (*Figure 2—figure supplement 1D*), an important control considering the debate regarding the permeability of this highly charged metabolite through the cell membrane (*Alva et al., 2016*). Interestingly, the effect of FBP appear most pronounced in the posterior, most undifferentiated PSM cells, while segmentation clock activity persists in the anterior PSM cells upon medium-supplementation of FBP (*Figure 2A*). This argues against a pleiotropic, toxic effect of FBP and suggests a more specific effect triggered by increased FBP levels.

As a second, complementary approach to alter glycolytic flux and hence FBP levels, we aimed to increase the activity of Pfk, the rate limiting glycolytic enzyme, in a genetic manner (*Figure 3*). To this end, we generated conditional transgenic mice which overexpress cytoPFKFB3 in a Cre-dependent manner. We showed that cytoPFKFB3 overexpression was indeed effective in increasing glycolytic flux in PSM explants, with a two-fold increase in secreted lactate (*Figure 3B*). Such a strong activation of glycolysis has been shown to be difficult to achieve by overexpression of single, wild-type glycolytic proteins in mammalian cell lines (*Tanner et al., 2018*; *Yalcin et al., 2009*). Of note, GC-MS analysis showed that cytoPFKFB3 overexpression was effective in increasing intracellular FBP levels

(*Figure 3C*, *Figure 3D*). Because the extent of glycolytic activation by cytoPFKFB3 was dependent on glucose concentration in the culture media (*Figure 3B*), we can titrate the effects of cytoPFKFB3 over-expression by increasing glucose. Therefore, the *cytoPFKFB3* transgenic mouse line that we generated is a powerful, genetic mouse model to study the function of glycolysis and, more importantly, that of a sentinel glycolytic metabolite FBP, in various biological contexts.

Functionally, overexpression of cytoPFKFB3 led to impairment of PSM segmentation at 10 mM or higher glucose concentrations, while wild-type PSM developed properly, at least qualitatively, at this glucose concentration (*Figure 4*, *Figure 1—figure supplement 1*). The abnormal PSM development accompanied disruption of the segmentation clock activity and suppression of Wnt-target gene expression, while expression of FGF-target gene remained comparable to control. These phenotypes are reminiscent of our observation that intermediate levels (10 mM) of exogenous FBP suppressed mRNA expression of *Msgn1* but not of *Dusp4* (*Figure 2B*). This data hence indicate that cytoPFKFB3 overexpression phenocopies the effect of the FBP-supplementation on PSM development.

Combined, these findings provide evidence that the sentinel glycolytic metabolite FBP exerts a signaling function in PSM development.

## The role of regulated flux at the level of Pfk

Our findings suggest that flux-regulation at the level of Pfk is critical to keep FBP steady state levels within a range compatible with proper PSM patterning and segmentation. In agreement with such a rate-limiting function for Pfk, we found in glucose titration experiments that FBP levels saturated and did not further increase at glucose levels above 10 mM (*Figure 3E*). Along similar lines, the supplementation of high concentrations of the Pfk substrate F6P did not result in a significant increase of FBP levels, again compatible with a rate-limiting function at the level of Pfk (*Figure 3E*). The upper limit of glycolytic flux and FBP levels can be experimentally increased by cytoPFKFB3 overexpression (*Figure 3B*, *Figure 3E*). We interpret the data as evidence that cytoPFKFB3 overexpression compromises the flux-control function of Pfk and hence much higher FBP (and secreted lactate) levels are reached. Such a drastic increase in glycolytic flux and FBP levels correlates with a severe PSM patterning phenotype (*Figure 4*), which resembles the phenotype induced by supplementation of high dose of FBP (*Figure 2*). Our results in mouse embryos hence provides evidence that flux regulation by Pfk, an evolutionary conserved role present from bacteria to humans, serves to maintain FBP levels below a critical threshold.

In addition to this threshold function, we find evidence that a change in glycolytic flux and FBP levels within the physiological range also correlates with functional consequences. For instance, we reveal flux-dependent quantitative gene expression changes, such as a control of Wnt-signaling target genes, during glucose titration experiments (*Figure 2—figure supplement 2*). Accordingly, a modest increase in glycolytic flux in *cytoPFKFB3* transgenic embryos cultured at 2.0 mM glucose also exhibits Wnt signaling target gene downregulation (*Figure 5A*). Of note, it remains unclear whether changes in the levels of FBP alone or if, in fact, changes in several sentinel metabolites underlies this gene expression change in *cytoPFKFB3* embryos (*Figure 3—figure supplement 1D*). Given the advent of technology detecting metabolite-protein interactions, including allosteric effects, in a proteome-wide manner (*Savitski et al., 2014*; *Feng et al., 2014*; *Piazza et al., 2018*), the fundamental challenge to reveal allosteromes for several metabolites should now be tackled.

## Wnt signaling as a link between glycolytic-flux and PSM patterning

While the detailed mechanism of flux-regulated gene expression in PSM cells has not yet been revealed, the response to changes in flux clearly involves the Wnt signaling pathway: lowering glucose concentration correlates with an upregulation of Wnt target genes, while the opposite effect was found when glucose concentration was increased (*Figure 2—figure supplement 2*). Consistent with such an anti-correlation, we found that Wnt target gene expression was decreased in conditions of FBP supplementation and cytoPFKFB3 overexpression (*Figure 2B*, *Figure 5*). Previously, it was shown that Wnt signaling can promote glycolysis directly or indirectly (*Oginuma et al., 2017*; *Pate et al., 2014*). Therefore, our findings suggest that in the PSM there is a negative-feedback regulation from glycolysis to Wnt signaling. Contrary to our findings, a previous study performed in cultured chick embryos has suggested that inhibition of glycolysis decreases Wnt signaling (*Oginuma et al., 2020*). This discrepancy could relate to the time point of analysis: while Oginuma et al. mainly focused on

analyzing samples 16 hr after metabolic changes, we chose to score the effects of altered glycolytic flux/FBP levels already after a 3-hr incubation, with the goal to capture the primary response of PSM cells. Whether the difference in sampling time underlies the observed difference is yet unknown, but both studies highlight that Wnt signaling is responsive to glycolytic flux, supporting a tight link between metabolism and PSM development. Given the central function of Wnt signaling in development, stem cells and disease, a future key interest will be to reveal its link to metabolism and in particular glycolytic flux in these different contexts. In addition, as FBP can be considered as a universal sentinel for glycolytic-flux in living organisms, it will be crucial to reveal the mechanisms of how cells integrate steady state FBP levels in these different contexts.

## Impact of altered glycolytic-flux and FBP levels on subcellular protein localization

As one mechanism by which FBP levels are integrated into cellular programs, we propose that FBP levels impact subcellular localization of proteins, some of which might function as FBP sensor molecules. Here, we revealed that several glycolytic enzymes including Aldoa and Pfkl are amongst those proteins altering their subcellular localization in response to FBP supplementation or cytoPFKFB3 overexpression in high glucose/flux conditions (*Figure 6*, *Figure 7*).

While we do not have any direct functional evidence so far for a functional role of nuclear localized glycolytic enzymes, our findings do raise the question whether their subcellular compartmentalization is linked to a non-metabolic, moonlighting function (*Enzo et al., 2015*; *Cieśla et al., 2014*; *Ronai et al., 1992*; *Yang et al., 2011*; *Yang et al., 2012*).

Additionally recent evidence in several biological systems highlights that subsets of metabolic reactions, for instance, from the mitochondrial TCA-cycle, take place also in the nucleus in order to maintain local supply of substrates for epigenetic modifications (*Nagaraj et al., 2017*; *Kafkia et al., 2020*). Thus, one possibility is that specific glycolytic reactions are taking place also in the nucleus, for instance to provide a local source of co-factors (*e.g.* NAD$^+$) and/or substrates (*e.g.* acetyl-CoA, O-GlcNAc) for post-translational modifications of proteins. This emerging view of compartmentalized, local metabolic reactions as a way to regulate cellular functions has been recently supported by experimental evidence (*De Bock et al., 2013*; *Hu et al., 2016*; *Jang et al., 2016*; *Ryu et al., 2018*; *Bulusu et al., 2017*).

While future studies will need to reveal if nuclear localization of glycolytic enzymes is linked to their moonlighting functions or metabolic compartmentalization, our finding that their subcellular localization is glycolytic flux-sensitive reveals a potentially general mechanism of how metabolic state is integrated into cellular programs. Of note, the translocation of proteins was observed only when high levels of FBP were reached upon direct FBP supplementation or cytoPFKFB3 overexpression with high glucose (*Figure 6*, *Figure 7*). Future studies hence need to investigate whether flux-dependent change in protein localization also occurs upon moderate and more physiological changes in glycolytic-flux/FBP levels. To this end, the development of more quantitative approaches, such as live-imaging of tagged enzymes and the development of metabolite biosensors, are needed.

## Outlook

Using mouse embryo mesoderm development as a model system, our study identifies FBP as a sentinel, flux-signaling metabolite connecting glycolysis and developmental signaling pathways. Interestingly, FBP has been implicated as an allosteric regulatorregulation of a multitude of proteins involved in either metabolic as well as non-metabolic processes in *Escherichia coli* (*Feng et al., 2014*; *Piazza et al., 2018*). Revealing the FBP allosterome and investigating the impact of allosteric interactions on protein localization and, more generally, on protein function, is of central importance and a key future objective (*Lindsley and Rutter, 2006*). Excitingly, emerging techniques now start to enable a more comprehensive interrogation of metabolite-protein interaction (*Savitski et al., 2014*; *Feng et al., 2014*; *Piazza et al., 2018*). We are currently exploring the possibility to decipher metabolite-protein allosteromes in complex biological samples, such as in developing embryos.

Lastly, it is notable that the role of FBP as a flux-signaling metabolite has been demonstrated in microbes (*Litsios et al., 2018*) and hence predates the origin of signaling pathways involved in multicellular organism development, such as the Wnt signaling pathway, which appeared in metazoa (*Holstein, 2012*). It is hence of great interest to investigate how metabolic flux-signaling has been

integrated into signaling pathways involved in multicellular organism development in the course of evolution.

## Materials and methods

### Mice

All animals were housed in the EMBL animal facility under veterinarians' supervision and were treated following the guidelines of the European Commission, revised directive 2010/63/EU and AVMA guidelines 2007. All the animal experiments were approved by the EMBL Institutional Animal Care and Use Committee (project code: 21–001_HD_AA). The detection of a vaginal plug was designated as embryonic day (E) 0.5, and all experiments were conducted with E10.5 embryos.

### Generation of conditional *cytoPFKFB3* transgenic mouse line

Flag-PFKFB3(K472A/K473A) (hereafter termed as cytoPFKFB3) from *Yalcin et al., 2009* was amplified by PCR using the following primers: Forward 5'-TAGGCCGGCCGCCACCATGGACTACAAGGACG ACGACG-3' and reverse 5'-TGGGCCGGCCGGAAATGGAATGGAACCGACAC-3'. The resulting amplicon was then cloned into the Rosa26 targeting vector Ai9 (*Madisen et al., 2010*) using *F*seI restriction enzyme to generate the loxP-stop-loxP-cytoPFKFB3 (LSL-cytoPFKFB3) construct. Conditional *cytoPFKFB3* transgenic mouse line was generated by standard gene targeting techniques using R1 embryonic stem cells. Briefly, chimeric mice were obtained by C57BL/6 blastocyst injection and then outbred to establish the line through germline transmission. $Rosa26^{LSL-cytoPFKFB3}$ mouse line was maintained by crossing to CD1 mouse strain.

### Genotyping

The following mice used in this study were described previously and were genotyped using primers described in these references: T-Cre (*Perantoni et al., 2005*), $Hprt^{Cre}$ (*Tang et al., 2002*), LuVeLu (*Aulehla et al., 2008*). Akita mice (*Wang et al., 1999*; *Yoshioka et al., 1997*) were imported from the Jackson Laboratory (stock #003548) and were genotyped using the following primers: Forward 5'-TGCTGATGCCCTGGCCTGCT-3' and reverse 5'-TGGTCCCACATATGCACATG-3' (restriction digestion of PCR products by *F*nu4HI produce 140 bp and 280 bp bands for wild-type and mutant alleles, respectively). The primers used for genotyping of $Rosa26^{LSL-cytoPFKFB3}$ mice were as follows: Bofore Cre-recombination, forward 5'-GAGCTGCAGTGGAGTAGGCG-3' and reverse 5'-CTCGACCATGGTAATA GCGA-3' (predicted product size, 580 bp); After Cre-recombination, forward 5'-GGCTTCTGGCGT GTGACCGG-3' and reverse 5'-ACTCGGCTCTGCGTCAGTTC-3' (predicted product size, 340 bp). For polymerase chain reaction (PCR), OneTaq 2 X Master Mix with Standard Buffer was utilized (New England Biolabs).

### Ex vivo culture of PSM explants

PSM explants with three intact somites were collected using micro scalpels (Feather Safety Razor, No. 715, 02.003.00.715) in DMEM/F12 (without glucose, pyruvate, glutamine, and phenol red; Cell Culture Technologies) supplemented with 0.5–25 mM glucose (Sigma-Aldrich, G8769), 2.0 mM glutamine (Sigma-Aldrich, G7513), 1.0% (w/v) BSA (Cohn fraction V; Equitech-Bio, BAC62), and 10 mM HEPES (Gibco, 15360–106). The explants were then washed with pre-equilibrated culture medium (DMEM/F12 supplemented with 0.5–25 mM glucose, 2.0 mM glutamine, and 1.0% (w/v) BSA) and were transferred to eight-well chamber slides (Lab-Tek, 155411) filled with 160 μl of the pre-equilibrated culture medium. When assessing the impacts of glycolytic intermediates on PSM development, culture medium supplemented with a glycolytic intermediate *i.e.* fructose 1-phosphate (Sigma-Aldrich, F1127), fructose 6-phosphate (Sigma-Aldrich, F3627), fructose 1,6-bisphosphate (Santa Cruz, sc-221476), $^{13}C_6$-fructose 1,6-bisphosphate (Cambridge Isotope laboratories, CLM-8962), 3-phosphoglycerate (Sigma-Aldrich, P8877) was prepared with pre-equilibrated culture medium right before dissection. Basal culture condition was 0.5 mM glucose at the beginning of this study but was later switched to 2.0 mM glucose which yields a slightly improved reporter gene expression. No major difference was observed in the effects of FBP between these glucose conditions. Following ex vivo culture under 5% $CO_2$, 60% $O_2$ condition, the explants were washed with PBS and were fixed overnight with 4% (v/v) formaldehyde solution (Merck, 1040031000) at 4 °C for further analyses.

## Time-lapse imaging of *LuVeLu* embryos

Imaging was performed as described before (*Lauschke et al., 2013*). In brief, samples were excited by 514 nm-wavelength argon laser or 960 nm-wavelength Ti:Sapphire laser (Chameleon-Ultra, Coherent) through 20×Plan-Apochromat objective (numerical aperture 0.8). In some experiments, samples were placed into agar wells (3% low Tm agarose, Biozyme, 840101) with 600 nm-width to restrain tissue movements during imaging. Image processing was done using the Fiji software (*Schindelin et al., 2012*).

## In situ hybridization

Fixed PSM explants were dehydrated with methanol and were stored at –20 °C until use. Whole mount in situ hybridization was performed as described in *Aulehla et al., 2008*.

## Immunostaining

Immunostaining with anti-cleaved caspase-3 antibody (Cell Signaling, #9661, RRID:AB_2341188; 1:200 dilution) was performed as described in *Bulusu et al., 2017*. Goat anti-rabbit-Alexa-488 antibody was used as a secondary antibody (Invitrogen, #A-11034; 1:1000 dilution). Samples were imaged on a LSM780 laser-scanning microscope (Zeiss) using 10×EC Plan-Neofluar objective lens (numerical aperture 0.3).

## Gas chromatography-mass spectrometry (GC-MS) analysis

Wild-type and *cytoPFKFB3* transgenic PSM explants with no somite were cultured ex vivo for three hours under different glucose conditions, as described above. After washing twice with ice-cold PBS, the explants were snap frozen by liquid $N_2$, and were stored at –80 °C until use. Metabolites were extracted from the 25 x explants by mechanically dissociating tissues by pipetting in 100 µl ice-cold methanol supplemented with ribitol (5.0 µg/mL) as an internal standard. For metabolite extraction from the conditioned medium, 20 µl of the medium was mixed with 40 µl of ice-cold methanol supplemented with ribitol. After incubation at 72 °C for 15 min, one volume of ice-cold MilliQ water was added, followed by centrifugation at 14,000 rpm at 4 °C for 10 min. The supernatants were transferred to amber glass vials (Agilent, 5183–2073) and were dried by centrifugal evaporator EZ-2 Plus (SP Scientific) (30 °C, Medium Boiling Point). The dried metabolite extracts were derivatized with 40 µL of 20 mg/mL methoxyamine hydrochloride (Alfa Aesar, 593-56-6) solution in pyridine (Sigma-Aldrich, 437611) for 90 min at 37 °C, followed by addition of 80 µL N-methyl-trimethylsilyl-trifluoroacetamide (MSTFA) (Alfa Aesar, 24589-78-4) and 10 hour incubation at room temperature (*Kanani and Klapa, 2007*; *Blasche et al., 2021*). GC-MS analysis was performed using a Shimadzu TQ8040 GC-(triple quadrupole) MS system (Shimadzu Corp.) equipped with a 30mx0.25 mm x 0.25 µm ZB-50 capillary column (7HG-G004-11; Phenomenex). One µL of the sample was injected in split mode (split ratio = 1:5) at 250 °C using helium as a carrier gas with a flow rate of 1 mL/min. GC oven temperature was held at 100 °C for 4 min followed by an increase to 320 °C with a rate of 10 °C/min, and a final constant temperature period at 320 °C for 11 min. The interface and the ion source were held at 280°C and 230°C, respectively. The detector was operated both in scanning mode (recording in the range of 50–600 m/z) as well as in MRM mode (for specified metabolites). For peak annotation, the GCMSsolution software (Shimadzu Corp.) was utilized. The metabolite identification was based on an in-house database with analytical standards utilized to define the retention time, the mass spectrum and marker ion fragments for all the quantified metabolites. The metabolite quantification was carried out by integrating the area under the curve of the MRM transition of each metabolite. The data were further normalized to the area under the curve of the MRM transition of ribitol.

## Liquid chromatography-mass spectrometry (LC-MS) analysis

After three-hour culture in the presence of 20 mM $^{13}C_6$-FBP, PSM explants were washed with cold 154 mM ammonium acetate, snap frozen in liquid $N_2$ and then dissociated in 0.5 mL ice-cold methanol/water/ACN (50:20:30, v/v) containing 0.20 µM of the internal standard lamivudine (Sigma-Aldrich, PHR1365). The resulting suspension was transferred to a reaction tube, mixed vigorously and centrifuged for 2 min at 16,000×g. Supernatants were transferred to a Strata C18-E column (Phenomenex, 8B-S001-DAK) which were previously activated with 1 mL of $CH_3CN$ and equilibrated with 1 mL of $MeOH/H_2O$ (80:20, v/v). The eluate was dried in a vacuum concentrator. The dried

metabolite extracts was dissolved in 50 µL 5 mM NH$_4$OAc in CH$_3$CN/H$_2$O (75:25, v/v), and 3 µL of each sample was applied to an amide-HILIC (2.6 µm, 2.1x100 mm, Thermo Fisher, 16726–012105). Metabolites were separated at 30 °C by LC using a DIONEX Ultimate 3000 UPLC system and the following solvents: solvent A consisting of 5 mM NH$_4$OAc in CH$_3$CN/H$_2$O (5:95, v/v) and solvent B consisting of 5 mM NH$_4$OAc in CH$_3$CN/H$_2$O (95:5, v/v). The LC gradient program was: 98% solvent B for 1 min, followed by a linear decrease to 40% solvent B within 5 min, then maintaining 40% solvent B for 13 min, then returning to 98% solvent B in 1 min and then maintaining 98% solvent B for 5 min for column equilibration before each injection. The flow rate was maintained at 350 µL/min. The eluent was directed to the hESI source of the Q Exactive mass spectrometer (QE-MS; Thermo Fisher Scientific) from 1.85 min to 18.0 min after sample injection. The scan range was set to 69.0–550 m/z with a resolution of 70,000 and polarity switching (negative and positive ionisation). Peaks corresponding to the calculated metabolites masses taken from an in-house metabolite library (MIM +/− H$^+$ ±2 mmU) were integrated using the El-MAVEN software (*Melamud et al., 2010*). For the targeted quantification of FBP, extraction was performed as stated above with the following exceptions: samples (25 PSM) were dissociated in 0.5 mL ice-cold methanol/water/ACN (50:20:30, v/v) containing 0.25 uM U-13C6 FBP (Cambridge isotope laboratories, CLM-8962). After drying the samples were dissolved in 30 uL 5 mM NH$_4$OAc in CH$_3$CN/H$_2$O (75:25, v/v), and 13 µL of each sample was applied to the amide-HILIC column. The LC gradient program was: 98% solvent B for 2 min, followed by a linear decrease to 30% solvent B within 3 min, then maintaining 30% solvent B for 15 min, then returning to 98% solvent B in 1 min and then maintaining 98% solvent B for 5 min for column equilibration before each injection. The scan range was set to 200–500 m/z with a resolution of 70,000 and only done in negative mode.

## Extracellular lactate measurement

Condition medium was collected following 12 hr ex vivo culture of PSM explants, and was stored at –80 °C until use. Fluorometric lactate measurements were performed with the Lactate Assay Kit (Biovision, K607) following manufacturer's instructions with a slight modification. The reaction volume was reduced to 50 µl, and 0.5–1.0 µl of the conditioned medium was used for the analysis.

## Whole embryo roller-culture and TUNEL staining

Embryos were collected with the intact yolk sac at E8.5 in DMEM (1.0 g/L glucose, without glutamine and phenol red) (Gibco, 11880–028) supplemented with 2.0 mM glutamine, 10%(v/v) FCS, and 1%(v/v) penicillin/streptomycin (Gibco, 15140–122). The embryos were cultured for 24 hours using the roller bottle culture system in 50% rat serum/DMEM (supplemented with 2.0 mM glutamine and 1% (v/v) penicillin/streptomycin) under 8% CO$_2$, 20% O$_2$, and 72% N$_2$ (flow rate, 20 mL/min) condition (*Rivera-Pérez et al., 2010*). Following the whole embryo culture, the embryos without the yolk sac and amniotic membrane were fixed with 4% formaldehyde overnight at 4 °C. TUNEL staining was done with In Situ Cell Death Detection Kit (Roche, 12156792910) following manufacturer's instructions, followed by DAPI (0.5 µg/mL) staining. Images were acquired with a LSM780 laser-scanning microscope (Zeiss) using 10×EC Plan-Neofluar objective lens (numerical aperture 0.3).

## Subcellular proteome analysis by mass spectrometry

PSM explants (without somites) were cultured in desired culture conditions. The explants were washed twice with ice-cold PBS and subjected to subcellular protein extraction using a Subcellular Protein Fractionation for Cultured Cells kit (Thermo Fisher Scientific, #78840). 8–11 x PSMs were used for each condition in each replicate. PSMs were dissociated in 10 µl of CEB buffer per PSM by pipetting, after which 10 µl (*i.e.* 1×PSM worth) of uncleared lysate was taken as the whole-cell lysate (WCL) sample. The rest of the extraction was carried out following manufacturer's instructions using buffer amounts scaled according to the number of PSMs in the sample. When using cytoPFKFB3 and control explants, subcellular protein extraction was performed with the following exceptions: After extraction of the MEM fraction, protein from the remaining pellet (constituting the nuclear and cytoskeletal fractions) was extracted with the NEB buffer (with micrococcal nuclease) plus 1×SDS lysis buffer [50 mM HEPES-NaOH (pH 8.5), 1% SDS, 1 x cOmplete protease inhibitor cocktail (Roche, 11873580001)]. The resulting fractions were stored at –80 °C before further processing. Subsequently, CYT and MEM fractions were reduced in volume to ~50 µl in a speedvac, and each subcellular protein fraction was denatured with 1% SDS at 95 °C for 5 min, after which residual nucleic acids were degraded with

benzonase (EMD Millipore, #71206-25KUN; final concentration 0.1–1 U/µl) for 45 min at 37 °C and 300 rpm until samples were no longer viscous.

All samples were prepared for MS using a modified SP3 protocol (Hughes et al., 2014). Briefly, protein samples were precipitated onto Sera-Mag SpeedBeads (GE Healthcare, #45152105050250 and #65152105050250) in the presence of 50% ethanol and 2.5% formic acid (FA) for 15 min at room temperature, followed by four washes with 70% ethanol on magnets. Proteins were digested on beads with trypsin and Lys-C (5 ng/µl final concentration each) in 90 mM HEPES (pH 8.5), 5 mM chloroacetic acid and 1.25 mM TCEP overnight at room temperature shaking at 500 rpm. Peptides were eluted on magnets using 2% DMSO and dried in a speedvac. Dry peptides were reconstituted in 10 µl water and labelled by adding 4 µl TMT label (20 µg/µl in acetonitrile (ACN)) (TMT10plex, Thermo Fisher Scientific #90110, comparison of FBP and F6P treatment; or TMTsixplex, #1861431, comparison of TG to Ctrl) and incubating for one hour at room temperature. Samples were multiplexed as follows: for comparison of FBP and F6P treatment all conditions (FBP, F6P, untreated) of each biological replicate were run in two separate TMT sets: one set including WCL, CYT and NUC fractions, and the other MEM, CHR and SKEL, for a total of six TMT sets for the experiment. For the comparison of Tg and Ctrl, both conditions (Tg, Ctrl) and all replicates were run in a single TMTsixplex experiment for each subcellular fraction (nuclear-cytoskeletal, cytoplasmic, membrane). Labeling was quenched with hydroxylamine (1.1% final concentration), and samples were dried in a speedvac. Each sample was then resuspended using 100 µl LC-MS $H_2O$, and 10% of each sample was taken, pooled to make a full TMT set, and desalted on an OASIS HLB µElution plate (Waters 186001828BA); washing twice with 0.05% FA, eluting with 80% ACN, 0.05% FA, and drying in a speedvac. The resulting sample was run on a 60 min LC-MS/MS gradient (see details below) to estimate relative amounts of protein in each channel. A second TMT set was then pooled using equalized amounts based on the median intensity of each channel from the first run to create an analytical TMT set with approximately equal labelled input protein in each channel. The analytical TMT set peptides were desalted using OASIS as described above and dried in a speedvac. Dried peptides were taken up in 20 mM ammonium formate (pH 10) and prefractionated offline into six (comparison of FBP, F6P to untreated) or 12 (comparison of Tg to Ctrl) fractions on an Ultimate 3000 (Dionex) HPLC using high-pH reversed-phase chromatography (running buffer A: 20 mM ammonium formate pH 10; elution buffer B: ACN) on an X-bridge column (2.1x10 mm, C18, 3.5 µm, Waters). Prefractionated peptides were vacuum dried.

For LC-MS/MS analysis, peptides were reconstituted in 0.1% FA, 4% ACN and analyzed by nanoLC-MS/MS on an Ultimate 3000 RSLC (Thermo Fisher Scientific) connected to a Fusion Lumos Tribrid (Thermo Fisher Scientific) mass spectrometer, using an Acclaim C18 PepMap 100 trapping cartridge (5 µm, 300 µm i.d. x 5 mm, 100 Å) (Thermo Fisher Scientific) and a nanoEase M/Z HSS C18 T3 (100 Å, 1.8 µm, 75 µm x 250 mm) analytical column (Waters). Solvent A: aqueous 0.1% FA; Solvent B: 0.1% FA in ACN (all LC-MS grade solvents are from Thermo Fisher Scientific). Peptides were loaded on the trapping cartridge using solvent A for 3 min with a flow of 30 µl/min. Peptides were separated on the analytical column with a constant flow of 0.3 µl/min applying a 120 min gradient of 2–40% of solvent B in solvent A. Peptides were directly analyzed in positive ion mode with a spray voltage of 2.2 kV and a ion transfer tube temperature of 275 °C. Full scan MS spectra with a mass range of 375–1500 m/z were acquired on the orbitrap using a resolution of 120,000 with a maximum injection time of 50ms. Data-dependent acquisition was used with a maximum cycle time of 3 s. Precursors were isolated on the quadrupole with an intensity threshold of 2e5, charge state filter of 2–7, isolation window of 0.7 m/z. Precursors were fragmented using HCD at 38% collision energy, and MS/MS spectra were acquired on the orbitrap with a resolution of 30 000, maximum injection time of 54ms, normalized AGC target of 200%, with a dynamic exclusion window of 60 s.

The proteomics data have been deposited to the ProteomeXchange Consortium via the PRIDE (Perez-Riverol et al., 2019) partner repository with the dataset identifier PXD029988. Mass spectrometry raw files were processed using IsobarQuant (Franken et al., 2015) and peptide and protein identification was obtained with Mascot 2.5.1 (Matrix Science) using a reference mouse proteome (uniprot Proteome ID: UP000000589, downloaded 14.5.2016) modified to include known common contaminants and reversed protein sequences. Mascot search parameters were: trypsin; max. 2 missed cleavages; peptide tolerance 10 ppm; MS/MS tolerance 0.02 Da; fixed modifications: Carbamidomethyl (C), TMT10plex (K); variable modifications: Acetyl (Protein N-term), Oxidation (M), TMT10plex (N-term). IsobarQuant output data was analyzed on a protein level in R using in-house

data analysis pipelines. In brief, protein data was filtered to remove contaminants, proteins with less than 2 unique quantified peptide matches as well as proteins, which were only detected in a single replicate. Subsequently, protein reporter signal sums were normalized within each TMT set using the vsn package (*Huber et al., 2002*). Significantly changing proteins between the treated and untreated sample were identified by applying a limma analysis (*Ritchie et al., 2015*) on the vsn-corrected values. Replicates were treated as covariates in the limma analysis for the comparison of FBP to F6P, as biological replicates were run as separate TMT sets. Multiple-testing adjustment of *p* values was done using the Benjamini-Hochberg method.

## Western blot analysis

PSM explants (without somites) were cultured in desired culture conditions and were subjected to subcellular protein extraction, as described above. Primary antibodies used in the study are as follows: Anti-Aldolase A (Proteintech, 11217–1-AP, RRID:AB_2224626, 1:5000), anti-Tpi (Acris, AP16324PU-N, RRID:AB_1928285, 1:5000), anti-Gapdh (Millipore, MAB374, RRID:AB_2107445, 1:5,000), anti-Pkm1/2 (Cell signaling, 3190, RRID:AB_2163695, 1:5000), anti-Histone H2B (Millipore, 07–371, RRID:AB_310561, 1:10,000), anti-beta-Tubulin (Millipore, 05–661, RRID:AB_309885, 1:10,000), anti-Hsp90 (Cell signaling, 4874, RRID:AB_2121214, 1:1000). Mouse monoclonal antibody against Pfkl was generated by EMBL Monoclonal Antibody Core Facility using full-length Pfkl as an antigen. For protein expression and purification, full-length Pfkl transcript was amplfied by reverse transcription (RT)–PCR using mouse embryo total RNA as a template and cloned into pET28M-SUMO3 vector (EMBL Protein Expression and Purification Core Facility) using *Age*I and *Not*I restriction enzymes. Following primers were used for RT-PCR: forward 5′-TCATCTACCGGTGGAATGGCTACCGTGGACCTGGAGA-3′ and reverse 5′-TCATCTGCGGCCGCTCAGAAACCCTTGTCTATGCTCAAGGT-3′.

## Gene expression analysis by NanoString nCounter analysis system

A custom probe set was designed to include 237 genes involved in glucose metabolism, Notch-, Wnt-, and FGF-signaling pathways. In addition, six positive controls, eight negative controls and housekeeping genes for normalisation (housekeeping genes used: *Cltc*, *Gusb*, *Hprt1* and *Tubb5*) were included in the probe set. Following three-hour culture with the specified glucose concentration, the PSM explants were further dissected immediately posterior to the neural tube to isolate the posterior PSM. Five posterior PSM samples were pooled per replicate and snap frozen by liquid $N_2$. Total RNA was isolated using TRIzol reagent (Invitrogen) according to manufacturer's instructions and concentrated using RNA Clean & Concentrator-5 Kit (Zymo research). RNA was hybridized to the probes at 65 °C, samples were inserted into the nCounter Prep Station for 3 hr, the sample cartridge was transferred to the nCounter Digital Analyzer, and counts were determined for each target molecule. Counts were analysed using nSolver Analysis Software Version 4.0, and sequentially subjected to background correction, positive control (quality control) and normalisation to housekeeping genes.

## RNA sequencing analysis

Libraries for RNA sequencing analysis were prepared following the Smart-seq2 protocol with small modifications (*Picelli et al., 2014*). Following three hour incubation, tail buds (posterior to the end of neural tube) were isolated from the PSM explants, washed twice with cold PBS (0.01% BSA), and mechanically dissociated in cold PBS (0.01% BSA) in micro wells (ibidi, #80486; one tail bud in 2 µl PBS). Cell suspensions (1.75 µl) were mixed with cell lysis buffer (4.25 µl; 0.02% Triton-X with RNasin), snap frozen by liquid $N_2$, and stored at –80 °C until cDNA synthesis.

cDNAs were synthesized using SuperScript IV Reverse Transcriptase (Thermo Fisher Scientific) and amplified by PCR (9 cycles) with HiFi Kapa Hot start ReadyMix (Kapa Biosystems, KK2601). After clean-up with SPRI beads, concentrations of cDNA (50–9000 bp) samples were determined by the Bioanalyzer (Agilent, High Sensitivity DNA kit). A total of 250 pg cDNAs were then used for tagmentation-based library preparation. Size distribution and concentrations of the libraries were determined by the Bioanalyzer (Agilent, High Sensitivity DNA kit) and the Qubit Fluorometer (dsDNA High Sensitivity Kit), respectively. Twelve multiplexed libraries were sequenced in one lane using NextSeq 500 (Illumina) with 75 bp single-end readings. Sequencing reads were aligned to *Mus musculus* genome (GRCm38) with the STAR aligner (version 2.7.1 a) (*Dobin et al., 2013*). For Pfkfb3, reads from wild-type and mutant (cytoPFKFB3) transcripts were counted separately and summed up

as Pfkfb3 read counts. Differential gene expression analysis was performed with the DEseq2 package (*Love et al., 2014*) using gene count tables produced during alignment with GRCm38.101 annotation. Orphan genes with no gene symbol were excluded from the downstream analysis. Gene ontology (GO) term analysis was performed with DAVID. All RNA sequencing data used for this study have been deposited to the European Nucleotide Archive (ENA) under the accession number PRJEB55095.

## Statistical analysis

Statistical analysis was performed with GraphPad PRISM 9 software. For the metabolomics data, statistical analysis was performed with the Statistical Analysis for Microarray (SAM) package (*Tusher et al., 2001*) using R. For Pearson correlation analysis, numpy (*Harris et al., 2020*), pandas (*McKinney, 2010*), and scipy (*Virtanen et al., 2020*) libraries were used. For data visualization, matplotlib (*Hunter, 2007*) library was utilized.

## Acknowledgements

We thank Theodore Alexandrov for helpful discussion and Jonathan Rodenfels and Joel I Perez-Perri for critical comments on the manuscript. We thank Jason Chesney for kindly sharing the Flag-PFKFB3(K472A/K473A) plasmid, Yvonne Petersen for performing blastocyst injection of ESCs, Jana Kress for generating anti-Pfkl antibody, Vladimir Benes and Laura Villacorta for technical advice and supports for RNA-sequencing analysis, Jonathan Landry for helping RNAseq data analysis, the nCounter core facility at the university of Heidelberg for expression analysis using NanoString technology, Bernd Klaus for helping with statistical analysis of the NanoString data and all the member of the Aulehla group for their support and helpful discussion. This work was supported by the EMBL Advanced Light Microscopy Facility (ALMF), Genomics Core Facility and the Laboratory Animal Resources (LAR).

## Additional information

### Funding

| Funder | Grant reference number | Author |
|---|---|---|
| European Molecular Biology Laboratory | | Kiran R Patil<br>Martin Beck<br>Alexander Aulehla |
| H2020 Marie Skłodowska-Curie Actions | 664726 | Hidenobu Miyazawa |
| Japan Society for the Promotion of Science | | Hidenobu Miyazawa |
| Sigrid Juséliuksen Säätiö | | Henrik M Hammarén |
| Deutsche Forschungsgemeinschaft | 331351713 | Alexander Aulehla |

The funders had no role in study design, data collection and interpretation, or the decision to submit the work for publication.

### Author contributions

Hidenobu Miyazawa, Conceptualization, Investigation, Writing - original draft, Writing – review and editing; Marteinn T Snaebjornsson, Conceptualization, Investigation, Writing – review and editing; Nicole Prior, Conceptualization, Investigation; Eleni Kafkia, Henrik M Hammarén, Formal analysis, Investigation; Nobuko Tsuchida-Straeten, Resources; Kiran R Patil, Martin Beck, Supervision, Project administration; Alexander Aulehla, Conceptualization, Supervision, Funding acquisition, Project administration, Writing – review and editing

### Author ORCIDs

Hidenobu Miyazawa 🔟 http://orcid.org/0000-0001-6164-5134
Nicole Prior 🔟 http://orcid.org/0000-0003-2856-7052

Henrik M Hammarén [ID] http://orcid.org/0000-0002-8534-2530
Kiran R Patil [ID] http://orcid.org/0000-0002-6166-8640
Martin Beck [ID] http://orcid.org/0000-0002-7397-1321
Alexander Aulehla [ID] http://orcid.org/0000-0003-3487-9239

## Ethics

All animals were housed in the EMBL animal facility under veterinarians' supervision and were treated following the guidelines of the European Commission, revised directive 2010/63/EU and AVMA guidelines 2007. All the animal experiments were approved by the EMBL Institutional Animal Care and Use Committee (project code: 21-001_HD_AA).

## Decision letter and Author response

Decision letter https://doi.org/10.7554/eLife.83299.sa1
Author response https://doi.org/10.7554/eLife.83299.sa2

## Additional files

### Supplementary files

• Supplementary file 1. The list of DEGs between control and cytoPFKFB3 explants under 10 mM glucose condition.
• Supplementary file 2. GO term analysis with the DEGs identified in cytoPFKFB3 explants.
• Supplementary file 3. The list of DEGs between control and FBP-treated explants.
• Supplementary file 4. GO term analysis with the DEGs identified in FBP-treated explants.
• Supplementary file 5. The list of DEGs between control and FBP-treated explants.
• Supplementary file 6. GO term analysis with the DEGs identified in F6P-treated explants.
• MDAR checklist

### Data availability

RNAseq data have been deposited to the European Nucleotide Archive (ENA) under the accession number PRJEB55095. Proteomics data have been deposited to the ProteomeXchange Consortium under the accession number PXD029988.

The following datasets were generated:

| Author(s) | Year | Dataset title | Dataset URL | Database and Identifier |
|---|---|---|---|---|
| Miyazawa H, Snaebjornsson MT, Prior N, Kafkia E, Hammarén HM, Tsuchida-Straeten N, Patil KR, Beck M, Aulehla A | 2022 | Glycolytic flux-signaling in mouse embryos | https://www.ebi.ac.uk/ena/browser/view/PRJEB55095 | European Nucleotide Archive, PRJEB55095 |
| Miyazawa H, Snaebjornsson MT, Prior N, Kafkia E, Hammarén HM, Tsuchida-Straeten N, Patil KR, Beck M, Aulehla A | 2022 | Subcellular proteomics of murine presomitic mesoderm | http://proteomecentral.proteomexchange.org/cgi/GetDataset?ID=PXD029988 | ProteomeXchange, PXD029988 |

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
