## [Editor Report]

How can generic changes in metabolism program specific changes in signaling and cell fate? To date it has been difficult to distill plausible models for specificity in this arena, but now Miyazawa et al. demonstrate a link between glycolytic flux, mesoderm segmentation and Wnt signaling. Enhanced flux translated into failures in segmentation and suppression of Wnt target gene expression, as well as inducing alterations to subcellular localization of glycolytic enzymes, suggesting pivotal links between glycolytic flux, signaling and lineage specification. Through careful work on presomitic mesoderm, the authors work suggests important new links between metabolism and differentiation.

---

## [Decision Letter]

[Editors' note: this paper was reviewed by Review Commons.]

---

## [Author Response]

Reviewer #1 (Evidence, reproducibility and clarity (Required)):The paper by Miyazawa and colleagues addresses a key question: How is changed metabolic activity sensed and to induce changes in developmental programs. In recent years, there is more and more indication that metabolism is not only a dull workhorse synthesizing the building blocks for new cells and providing chemical energy, but that metabolic activity itself has also a regulatory role. How this precisely works is largely unknown and even also unexplored in higher cells. From early insights obtained in microbes, it seems that certain metabolites – possibly reflecting metabolic activity (i.e. flux) – could be metabolic signals that feedback into cellular regulation.The current paper takes this idea now to developmental processes, where the authors found that the glycolytic metabolite fructose-1,6-bisphosphate is a flux-dependent signal that interferes with developmental processes. This is a very exciting finding, as it indicates that this metabolite not only has a regulatory function in microbes but also in mouse during mesoderm development.Answering the question how such a flux-dependent metabolite mechanistically interferes with the developmental processes is an enormously difficult. Compared to other mechanistic studies, where deleting genes, modifying genes, and changing protein expressions will usually do the trick, here, perturbing metabolite levels is extremely challenging, particularly if such perturbations need to be carried out in a way that nothing else is perturbed. Researchers, who are not overly familiar with metabolism, usually underestimate the difficulty with targeted and insightful perturbation of metabolism.To this end, the authors of this paper need to be congratulated for a very well carried out study with very solid data, and excellent control experiments. The authors open up a new path towards understanding how embryo mesoderm development is regulated by metabolic activity. In particular, they show that glycolytic flux, FBP and important developmental phenotypes as well as protein localization changes are linked. As normal with a complex metabolism-based story as this one, there is always more that could be done. Yet, the results are highly important to be reported now such that the field as a whole can build on these interesting results and to explore the exciting path further that has been opened by the authors. Thus, I strongly recommend publishing these findings: The data generated by the authors are accompanied by the required control experiments. The conclusions drawn are very solid. I do not have any major concerns but just a number of minor suggestions that the authors could consider in a revised version of the manuscript.Minor:1. At the end of the introduction, the authors stated their original goal. As it is phrased, it is unclear whether this goal has been obtained or not. They might want to consider replacing the last introductory sentence by a sentence stating what the reader can find in this paper.

#1. We agree with the reviewer and have rephrased accordingly (line 90–93):

“In this study, our goal was therefore to first determine in vivo sentinel metabolites during mouse embryo PSM development. We then combined genetic, metabolomic and proteomic approaches to investigate how altered glycolytic flux and metabolite levels impact developmental signaling and patterning processes.”

2. Data from Figure 3: If you plot the lactate secretion vs the FBP levels of the controls and the overexpression experiment, would the control and the overexpression data lie on one line (maybe if combined with the data shown in Figure 1A)?

#2. As the reviewer suggested, it is of great interest to check whether lactate secretion and FBP levels show a similar correlation in control and cytoPfkfb3 embryos, considering that cytoPfkfb3 overexpression lifts the upper limit of glycolytic capacity and FBP levels (revised Figure 3B, 3E). As the reviewer suggested, we plotted FBP levels against lactate secretion and fitted a linear regression line onto control samples (please see Figure 3—figure supplement 1C ). The new plot shows that lactate secretion and FBP levels in cytoPfkfb3 embryos lie on the linear regression line derived from wild-type samples, highlighting that a correlation between lactate secretion and FBP levels is maintained even in cytoPfkfb3 embryos. We now included this new plot in the revised Figure 3—figure supplement 1 and modified the text accordingly (line 326-328):

“In addition, FBP levels showed a linear correlation with lactate secretion in control explants, and such a correlation was maintained even in cytoPfkfb3 explants (Figure 3—figure supplement 1C).”

3. Maybe the authors could attempt an experiment like the following one: Chose the strongest phenotype observed and test a combination of overexpressing cytoPfkfb3 and reducing extracellular glucose level at the same time?

#3. We agree this suggested experiment is important to show that the phenotype in cytoPfkfb3 embryos is indeed dependent on glycolytic flux and have already addressed this specific point in our manuscript, see results in Figure 4B and 5A in our original manuscript. The results show that the phenotypes in cytoPfkfb3 explants, i.e. reduction in somite formation and downregulation of *Msgn* mRNA expression occur in a glucose dose-dependent manner. Since in this embryonic context, we show that glucose concentration impacts glycolytic flux (see increased lactate production upon glucose titration in Figure 3B), our findings support the conclusion that the effect of cytoPfkfb3 overexpression is flux-dependent and not due to the overexpression per se. Based on the reviewer's feedback, we have modified the text to clarify and highlight this critical point (line 238–242):

“Combined, these results show that cytoPfkfb3 overexpression results in reduced segment formation, arrest of the segmentation clock oscillations and downregulation of Wnt signaling, in a glucose-dose dependent manner. As glucose concentration impacts, in turn, glycolytic flux (Figure 1A, 3B), these findings suggest that these phenotypes are flux-dependent and are not a mere result of cytoPfkfb3 overexpression.”

4. Can the proteomics experiments shown in Figure 6 be repeated with high and low extracellular glucose? High glucose should yield high FBP levels and one would then expect to see the same as with the experiment where at 2 mM glucose 20 mM extracellular FBP were added. Is this the case?

#4. We agree with the reviewer that based on the findings, one would expect the phenotype, i.e. in this case translocation of proteins, to correlate with FBP levels. Two of our results are of note in this regard.

First, our data indicates that in order to see the effect on protein localization, high levels of FBP have to be reached. Accordingly, we find that Pfkl becomes depleted from the nuclear-cytoskeletal fraction in cytoPfkfb3 explants when cultured in 10 mM glucose but not (visibly) in 2.0 mM glucose (Figure 7D). Corresponding to this, FBP levels in cytoPfkfb3 explants show a significant increase (about 3-fold) from 2.0 to 10 mM glucose conditions (revised Figure 3E).

Second, in control samples, FBP levels saturate in high glucose conditions. FBP levels in control samples do not further increase when glucose concentration is increased from 10mM to 25mM, and thus it does not become as high as in cytoPfkfb3 embryos cultured in 10 mM glucose (revised Figure 3E).

Therefore, in order to reveal the translocation, it requires an experimental strategy that leads to significantly increased FBP levels, such as in cytoPfkfb3 explants with high glucose condition, or alternatively, direct supplementation of FBP.

As also pointed out by the other reviewers, we are experimentally generating controlled conditions that exceed the physiological range which the embryo is exposed to. Accordingly, our data does not constitute evidence that under physiological conditions an alteration of protein localization in response to change in glycolytic flux and FBP levels occurs, at a smaller scale.

We regard our approach as a first step to reveal potential mechanisms and so far hidden possible responses to changes in metabolic flux. In order to see minor changes in translocation upon small changes in glycolytic-flux/FBP levels, more quantitative approaches, such as live-imaging of tagged proteins, will need to be developed. We hence decided to include these discussions in our revised manuscript (line 446-452):

“Of note, the translocation of proteins was observed only when high levels of FBP were reached upon direct FBP supplementation or cytoPfkfb3 overexpression with high glucose (Figure 6, 7). Future studies hence need to investigate whether flux-dependent change in protein localization occurs upon moderate and more physiological changes in glycolytic-flux/FBP levels. To this end, the development of more quantitative approaches, such as live-imaging of tagged enzymes and the development of metabolite biosensors, are needed.”

5. While the authors quantified proteins in different compartments, I was wondering whether they also looked for whole-embryo protein expression changes?

#5. We have not done protein expression analysis using whole embryos, or other isolated tissues in this study. This is indeed a potentially interesting future experimental comparison.

6. Throughout the manuscript, the authors state the glucose levels or cytoPfkfb3 changes the glycolytic flux. While I tend to agree with this, it is important to note that the authors have not directly measured glycolytic flux, but use the amount of accumulated lactate as a proxy. I think it is important to add this disclaimer at important points in the manuscript, such that readers are aware of this point.

#6. We fully agree with the reviewer and now have added the following sentence in the first result section to make this point clearer to the reader (line 99-100):

"Throughout this study, we used quantification of secreted lactate as a proxy for glycolytic flux due to the inability to directly measure flux in embryonic tissues."

7. Another aspect for changing FBP levels could be connected on what was found in yeast, where the FBP levels were found to oscillate with the cell cycle (https://pubmed.ncbi.nlm.nih.gov/31885198/). Could this be connected with the pattern formation here?

#7. This is indeed an interesting aspect to discuss; in the absence of experimental evidence connecting the observed pattern formation and cell cycle (though some classic work had suggested its existence) we have decided to omit the discussion of this potential link.

8. Line 606: The mentioned review article also covers yeast. As such, maybe the authors should replace the term "bacteria" with "microbes"?

#8. We modified our manuscript accordingly.

Reviewer #1 (Significance (Required)):**Referees cross-commenting**As I mentioned in my comment, targeted metabolic perturbations are extremely difficult. Perturbing a metabolite level without at the same time perturbing the flux through this pathways is difficult (of not impossible). Also, the opposite is the case.I am not sure whether experiments as the one suggested by reviewer 2 (comment 1) will really lead to results from which further conclusions can be drawn. Furthermore, there does not need to be a linear correlation between the extracellular glucose concentration and metabolic flux/FBP levels (as my reviewer colleague implies). Thus, I am not sure whether doing this experiment makes sense, or would lead to strengthened conclusions.Reviewer 2 also states "The lack of proven mechanism for the activity of FBP might restrict the real general impact of this work." I agree that we do not know the downstream targets of FBP, but finding them would likely require many years of additional work. Such work will not be initiated if this paper is not published, and it would be a pity if it would be further delayed. I feel that the evidence is strong enough that FBP has an important role and with this paper published, it will motivate others to look for the downstream targets.Reviewer 3 makes the point: "Given that FBP levels are highly correlated with extracellular glucose levels (which impact glycolytic flux )(TeSlaa and Teitell, 2014) the authors should elaborate on why progressive increase in extracellular glucose does not affect PSM patterning, in the same way that increasing FBP levels does. " Here, I feel my reviewer colleague might be overlooking that in biochemistry molecular interactions typically reach a saturation at some point. The correlation between extracellular glucose and glycolytic flux has likely only a range where these two measures linearly correlate. Similarily, the correlation between glycolytic flxu and FBP likely also exists only within a certain range, and finally FBP levels and the downstream targets likely also only linearly interact within bounds. Thus, the absence of a correlation at "extremes" does by no mean mean that what the authors propose is incorrect. In fact, it just shows what you expect from biomolecular interactions that there a limits to linear correlations.Reviewer #2 (Evidence, reproducibility and clarity (Required)):Summary.The work described in this paper first searches for potential sentinel metabolites of glycolytic flux, focusing on the process of somitogenesis during mouse embryonic development. By measuring the levels of different metabolites in the presomitic mesoderm (PSM) of E10.5 mouse embryos cultured in the presence of three different glucose concentrations, the authors identify 14 metabolites whose concentration rises with increasing glucose concentration in the culture medium. Among them, they selected fructose 1,6-bisphosphate (FBP) for further analyses, as it showed the highest linear correlation with extracellular glucose concentrations. They then show that addition of FBP to the incubation medium of cultured embryo tails interfere with somitogenesis and tail extension in a concentration-dependent fashion. In addition, they show that this effect is exacerbated when extracellular glucose levels are increased. By analyzing specific targets of Wnt and FGF signaling, the authors also show that addition of FBP down-regulates both signaling pathways in the PSM. They then use a genetic trick (ubiquitous overexpression of cytoPfkfb3) to increase FBP levels by allosteric activation of Pfk (the enzyme that produces FBP) in developing embryos. When tails from these transgenic embryos were cultured in vitro and exposed to various glucose concentrations somitogenesis was affected in a way resembling the effects of FBP on cultured tails from wild type embryos. The authors then go on to determine the subcellular localization of different proteins in tails incubated in the presence of various FBP concentrations to identify that some enzymes involved in the glycolytic pathway (and they specifically focus on Pfkl and Aldoa) are excluded from nuclear fractions at high FBP concentrations. The authors conclude that FBP functions as a flux-signaling metabolite connecting glycolysis and PSM patterning, potentially through modulating subcellular protein localization.Major commentsI think that in general the work described in this manuscript has been performed to the highest technical standards. However, I do not think that I can agree with the authors' conclusions (that FBP connects glycolysis with PSM patterning and that subcellular localization of glycolytic enzymes play a role in this process), which in my opinion go way beyond what can be proven by the data provided.1. Explants incubated with external glucose concentrations up to 25 mM have no obvious defects on somitogenesis or on the segmentation clock as determined by LuVeLu cycling activity. Under these conditions, explants are expected to contain very high FBP levels if this metabolite keeps its linear relationship with external glucose (in this work it was not measured beyond 10 mM glucose in the medium, where FBP concentration was already very high). This contrasts with the phenotypes observed upon exogenous supplementation of FBP, which affects somitogenesis already at 2 mM glucose. These latter results are at odds not only with the lack of phenotypic alterations under high glucose conditions, but also with the observation that exogenous addition of fructose 6-phosphate (F6P), the substrate of Pfk enzymes to generate FBP, does not alter somitogenesis. The authors take the absence of effects by incubation with F6P as a control of the specificity of FBP. However, as F6P is the natural substrate of Pfk, it is possible that supplementation of F6P also leads to an increase of FBP but in a way closer to a physiological condition. Therefore, I find it essential to determine FBP levels in tails incubated in the presence of increasing amounts of F6P, as if it increases FBP levels, similarly to what the authors described for the tails incubated with increasing glucose concentrations, it will have important implications to the interpretation of the work presented in this manuscript.

#9. We agree with the reviewer and to directly address this central point, we have performed an extended, additional experiment, collecting 375 embryos to quantify FBP levels under five conditions with three biological replicates.

There are two major results that we highlight here: First, we found that addition of F6P did not lead to increased FBP levels compared to control samples cultured in 10 mM glucose, which is in stark contrast to cytoPfkfb3 embryos cultured in 10 mM glucose (revised Figure 3E). Second, while increasing glucose concentration is mirrored by elevated FBP levels as we reported, we find clear evidence of saturation above a concentration of 10mM glucose: increasing glucose to 25mM does not increase FBP levels further (revised Figure 3E).

This saturation effect seen in glucose titration, but also the absence of elevated FBP upon F6P addition, might be expected outcomes because, as also the reviewer 1 pointed out in the response, Pfk is commonly considered to be a rate-limiting enzyme in the glycolytic pathway. We now have the direct experimental data supporting this hypothesis and thank the reviewers to have initiated this additional (very involved) experiment.

This new data allows us to conclude more firmly on the correlation between FBP levels and phenotype: at high FBP levels, which are seen in cytoPfkfb3 samples, we observe PSM patterning defects. These high levels are not reached even at 25mM glucose or upon F6P addition, due to the saturation at the level of PFK enzymatic step. Hence, while glucose titration does elevate FBP significantly until this saturation, FBP levels are not as high as in cytoPfkfb3 samples. As a correlative finding, we see that only those conditions with very high FBP levels, or the direct addition of high levels of FBP, cause the arrest of segmentation clock activity. At moderately elevated FBP levels, observed in control explants with high glucose or in cytoPfkfb3 explants with low glucose, clock activity continues and we find a quantitative effect at the level of gene expression, i.e. Wnt signaling target downregulation (Figure 2—figure supplement 2, Figure 5A).

The new data has been included in the revised manuscript and the text has been adjusted accordingly:

– (Result Part, line 178–184) "Consistently, we found that cytoPfkfb3 overexpression lifted the upper limit of FBP levels in PSM cells (Figure 3E, Figure 3—figure supplement 1B, Figure 3—figure supplement 1C). In control explants, FBP levels did not increase further when glucose concentration was increased from 10 mM to 25 mM. It was also the case when control explants were cultured in 20 mM of F6P (Figure 3E). These results indicate that the Pfk reaction carries a (rate-)limiting role for glycolytic flux and FBP levels, and that cytoPfkfb3 overexpression hinders the flux-regulation function of Pfk."

– (Discussion Part, line 378–391) “Our findings suggest that flux-regulation at the level of Pfk is critical to keep FBP steady state levels within a range compatible with proper PSM patterning and segmentation. In agreement with such a rate-limiting function for Pfk, we found in glucose titration experiments that FBP levels saturated and did not further increase at glucose levels above 10 mM (Figure 3E). Along similar lines, the supplementation of high concentrations of the Pfk substrate F6P did not result in a significant increase of FBP levels, again compatible with a rate-limiting function at the level of Pfk (Figure 3E). The upper limit of glycolytic flux and FBP levels can be experimentally increased by cytoPfkfb3 overexpression (Figure 3B, 3E). We interpret the data as evidence that cytoPfkfb3 overexpression compromises the flux-control function of Pfk and hence much higher FBP (and secreted lactate) levels are reached. Such a drastic increase in glycolytic flux and FBP levels correlates with a severe PSM patterning phenotype (Figure 4), which resembles the phenotype induced by supplementation of high dose of FBP (Figure 2). Our results in mouse embryos hence provides evidence that flux regulation by Pfk, an evolutionary conserved role present from bacteria to humans, serves to maintain FBP levels below a critical threshold.”

The main difference between the experiments involving FBP supplementation and those involving high glucose concentrations or exogenous F6P addition is that in the later two cases increase in FBP would be restricted to the tissue(s) expressing Pfk, whereas upon FBP supplementation this metabolite would hit any tissue, regardless of whether or not it would ever be physiologically exposed to this molecule. In the case of the PSM, this might be relevant because it has been shown that there is a gradient of glycolysis, being high at the caudal tip and becoming lower at more anterior regions of the PSM, most likely mirroring the distribution of Pfk activity. Exogenous administration of FBP would flatten the gradient, which could lead to alterations in PSM patterning, whereas glucose (and eventually F6P) would not as they would increase FBP locally in the area where it is normally activated, keeping the natural gradient.On the basis of these arguments, to which extent does FBP connect glycolysis and somitogenesis under physiological conditions?

#10. First, we would like to clarify that while indeed glycolytic activity is graded along the PSM, as other and we reported previously (reported in Bulusu et al., 2017 and Oginuma et al., 2017), the baseline expression of the entire glycolytic machinery (from glucose transport to lactate production) is very high, in all PSM cells. Hence, we see that cells all along the entire PSM have very active glycolysis, the posterior PSM being even more active.

For this and related reasons, our interpretation about the difference seen between glucose titration/F6P addition on one side, and FBP addition/cytoPfkfb3 addition on the other side, is based on the role of Pfk in controlling either flux levels or dynamics in all PSM cells.

Hence, while we agree that we generate experimental conditions that allow FBP levels to surpass those found in control embryos, we would like to highlight the fact that even moderate changes in flux does result in very robust functional consequences on gene expression (Figure 2—figure supplement 2, Figure 5), as we show in this work.

We can currently not fully address the first point raised, i.e. the role of graded flux/graded metabolite levels, due to the experimental limitations. Such a study requires, for instance, the generation of metabolite biosensor reporter lines in order to be able to monitor these changes dynamically, in space and time.

Essential additional experiment related to point #1: Measure FBP from PSM explants incubated under various exogenous concentrations of F6P.

#11. We have performed this suggested experiment, which required the collection of n=375 embryos cultured under the various conditions and analysis by LC-MS to quantify metabolites. The outcome was indeed very informative (please refer to our response #9).

Another experiment that could be informative: measure FBP levels in PSM incubated under different glucose concentrations but instead of using the whole PSM together, dividing the PSM in posterior, medium and anterior parts (similarly to what was done in Oginuma et al., 2017, reference in the manuscript) to see if there is a gradient in FBP activation.

#12. While in principle we agree that this experiment could be informative, we consider the proposed experiment beyond the scope of this work and technically very challenging (although possible). With a similar motivation, the development of metabolite biosensors is an alternative route that we are pursuing for future studies (for the detail, please refer to our response #10).

2. A similar argument could be presented for the results with the cytoPfkfb3 transgenics, as they are based on global artificial overactivation of Pfk, in addition to other possible effects of the ectopic activity of cytoPfkfb3, which were not controlled. Also, while the phenotypic alterations in the PSM in vitro, most particularly in the experiments involving incubation of the tails, are rather strong, the reported effects on somitogenesis in vivo are minor, also questioning the contribution of the in vitro conditions to the final phenotypic effects observed throughout the manuscript.

#13. First of all, we would like to emphasize that the phenotype seen in cytoPfkfb3 embryos, i.e. the reduction of segmentation and downregulation of Wnt-target gene expression, occurs in a glucose dose dependent manner (Figure 4B and 5A). Hence, it is not the overexpression of cytoPfkfb3 *per se* that can account for the effects seen. But rather, increased glycolytic flux caused by the combination of transgene expression with high glucose results in functional consequences.

In addition, ‘other possible effects’ that the reviewer is referring to should be evident in all transgenic embryos, irrespective of glucose dose. To the contrary, transgenic embryos cultured in low glucose conditions appear unaltered to control embryos.

Second, we agree that we need to distinguish between strong phenotypes, visible at the level of clock arrest, and milder phenotypes, visible at the level of quantitative gene expression changes. It is important to note that the moderate phenotype, i.e. the quantitative gene expression changes seen in posterior PSM, are seen upon the addition of FBP at moderate levels and upon in glucose titration within the physiological concentration range, as well as in cytoPfkfb3 embryos. We take this as evidence that the effects seen in cytoPfkfb3 transgenic embryos reflect a common response also seen under physiological conditions.

To extend this argument to the in vivo setting, we have performed additional experiments using a genetic mouse model for diabetes. As shown in our previous submission, cytoPfkfb3 transgenic animals do not exhibit a drastic in vivo phenotype when dissected at embryonic day 10.5. One interpretation of this finding is that since the cytoPfkfb3 phenotype is glucose and flux-dependent, the in vivo flux is low, reflecting low glucose concentrations described in vivo. To test the effect of increased flux in cytoPfkfb3 embryos in vivo, we therefore crossed the transgenic mice into a diabetic model called Akita, in which a point mutation in the Insulin2 gene causes high maternal glucose levels (Yoshioka et al., 1997; Wang et al., 1999). Using this experimental setup, we tested whether transgenic embryos in Akita diabetic females would manifest in vivo phenotypes.

Indeed, we found that cytoPfkfb3 transgenic embryos developing in Akita diabetic females showed significantly increased cases of neural tube closure defects (50% of cytoPfkfb3 embryos) and developmental delay (control: 38 somites vs. cytoPfkfb3: 34 somites at E10.5), defects not seen in transgenic cytoPfkfb3 embryos from control females (please refer to Figure 4—figure supplement 1D-E). This dependency of the in vivo phenotype on maternal glucose conditions again highlights that the defects observed in cytoPfkfb3 embryos are not due to the expression of cytoPfkfb3 per se, but are rather directly linked to increased/unregulated glycolytic flux.

We included the new in vivo data in the revised Figure 4—figure supplement 1D-E and modified the text accordingly.

In conclusion, combining the arguments in the two previous comments, to which extent the results from the addition of FBP or from the transgenic activation of Pfk are not artefactual phenotypes without real physiological relevance?

#14. In our view, two main conclusions, both in vivo and in vitro, can be drawn based on the result we obtained:

First, we find that a moderate increase in glycolytic flux, within the physiological range, leads to a quantitative and consistent change in gene expression, such as downregulation of Wnt target genes (Figure 2—figure supplement 2, Figure 5). Such a phenotype was the result of either glucose titration or culturing cytoPfkfb3-transgenic embryos in low glucose concentration.

In these conditions, while overall PSM patterning is qualitatively normal, we do find consistent changes at quantitative level, i.e. gene expression changes, which are also mirrored by a reduced rate of segmentation (Figure 4B). A detailed analysis of the quantitative changes at the level of segmentation clock dynamics is being carried out and will be presented in a dedicated follow up study.

Second, we find that a very significant increase in FBP levels, i.e. when cytoPfkfb3 transgenic animals are cultured in high glucose conditions or when samples are cultured in high levels of FBP, PSM patterning is qualitatively altered and segmentation clock ceases to oscillate. In this case, we agree that it is not a physiological condition, as such high levels of flux and FBP are not reached in control samples which have intact flux regulation by Pfk. Nevertheless, such an experimental condition can be insightful, as it very clearly reveals the potential link between glycolysis, clock activity, PSM patterning and the Wnt signaling pathway.

It is the combination between the moderate and the more severe effects, observed both in vitro, and now also in vivo using the Akita model (see above), that we take as evidence for an intrinsic, physiological link between glycolytic activity, PSM patterning and signaling.

3. The authors seem to give a strong functional meaning to the absence of Pfkl and Aldoa from the nuclear fraction in tails incubated with exogenous FBP, suggesting a "moonlighting" function of these enzymes under FBP regulation. In addition to the purely speculative nature of this interpretation (there is no proof for such activity or even an attempt to test it), the data provided is also difficult to interpret for various reasons.

#15. We fully agree that we do not show a functional role for either the nuclear localization of enzymes or their dynamic change in sub-cellular localization and have tried to express this clearly in the original manuscript:

– (Result Part, line 382-388) “While we have not been able to address the functional consequence of specific changes in subcellular localization, such as the nuclear depletion of Pfkl or Aldoa when glycolytic flux is increased, these results pave the way for future investigations on the mechanistic underpinning of how metabolic state is linked to cellular signaling and functions.”

– (Discussion Part, line 575-577): “While future studies will need to reveal if nuclear localization of glycolytic enzymes is linked to their moonlighting functions or metabolic compartmentalization…”

Based on this comment by the reviewer, we have further emphasised this point in the revised manuscript (line 430-433):

“While we do not have any direct functional evidence so far for a functional role of nuclear localized glycolytic enzymes, our findings do raise the question whether their subcellular compartmentalization is linked to a non-metabolic, moonlighting function.”

The protein levels in nuclear fractions are clearly much lower than those in the cytoplasm (this is best seen in the blots of Figure 6D). Does this represent similar subcellular distribution of these enzymes throughout the tissue or the different levels result from the presence of the enzymes in the nucleus of only a subset of the cells? This might be of importance to understand the possible relevance of the subcellular distribution of those enzymes. All the analyses were done on bulk tissue and, therefore, it is not possible to distinguishing between these possibilities. As the authors have antibodies for these enzymes, they could try to perform immunofluorescence analyses, which would provide spatial data.

#16: We agree that a spatially resolved analysis of the subcellular localization of these various enzymes is needed. Unfortunately, the immunofluorescence experiments that we performed did not yield clear, reliable results and hence we can’t provide the answer at this time.

In addition to this, it would be important to determine Pfkl and Aldoa subcellular localization in explants incubated with different external concentrations of glucose, which in a way reproduces better possible physiological effects (see point 1), to see if under those conditions high FBP also affects subcellular distribution of those enzymes.

#17: Please find our response under #4 (below), as this important point was also raised by the reviewer 1.

(Our response #4)

#4. We agree with the reviewer that based on the findings, one would expect the phenotype, i.e. in this case translocation of proteins, to correlate with FBP levels. Two of our results are of note in this regard.

First, our data indicates that in order to see the effect on protein localization, high levels of FBP have to be reached. Accordingly, we find that Pfkl becomes depleted from the nuclear-cytoskeletal fraction in cytoPfkfb3 explants when cultured in 10 mM glucose but not (visibly) in 2.0 mM glucose (Figure 7D). Corresponding to this, FBP levels in cytoPfkfb3 explants show a significant increase (about 3-fold) from 2.0 to 10 mM glucose conditions (revised Figure 3E).

Second, in control samples, FBP levels saturate in high glucose conditions. FBP levels in control samples do not further increase when glucose concentration is increased from 10mM to 25mM, and thus it does not become as high as in cytoPfkfb3 embryos cultured in 10 mM glucose (revised Figure 3E).

Therefore, in order to reveal the translocation, it requires an experimental strategy that leads to significantly increased FBP levels, such as in cytoPfkfb3 explants with high glucose condition, or alternatively, direct supplementation of FBP.

As also pointed out by the other reviewers, we are experimentally generating controlled conditions that exceed the physiological range which the embryo is exposed to. Accordingly, our data does not constitute evidence that under physiological conditions an alteration of protein localization in response to change in glycolytic flux and FBP levels occurs, at a smaller scale.

We regard our approach as a first step to reveal potential mechanisms and so far hidden possible responses to changes in metabolic flux. In order to see minor changes in translocation upon small changes in glycolytic-flux/FBP levels, more quantitative approaches, such as live-imaging of tagged proteins, will need to be developed. We hence decided to include these discussion in our revised manuscript (line 446-452):

“Of note, the translocation of proteins was observed only when high levels of FBP were reached upon direct FBP supplementation or cytoPfkfb3 overexpression with high glucose (Figure 6, 7). Future studies hence need to investigate whether flux-dependent change in protein localization occurs upon moderate and more physiological changes in glycolytic-flux/FBP levels. To this end, the development of more quantitative approaches, such as live-imaging of tagged enzymes and the development of metabolite biosensors, are needed.”

Suggested additional experiments related to point #3:3a. Analysis of subcellular localization of Pfkl and Aldoa by Immunofluorescence. This analysis is not limited by the amount of biological material available, so it could be applied to different experimental conditions.

#18. We addressed this point in our response #15.

3b. Subcellular distribution of Pfkl and Aldoa in explants exposed to different exogenous glucose concentrations. As this involves wild type embryos, it can be done following similar protocols as in figures 6 and 7 of the manuscript.

#19. We addressed this point in our response #16.

4. The results from the work presented in this manuscript would indirectly indicate a negative relationship between glycolysis and somitogenesis. This contrasts with previous reports indicating the essential role of aerobic glycolysis for the same process. There is no explanation for this apparent (and important) contradiction (the authors only comment the discrepancy between the data provided in this paper and previous reports in what concerns the relationship between glycolysis and Wnt signalling, although they also do not provide an explanation).

#19. We cannot resolve this discrepancy, but now offer a more detailed discussion, also based on the additional data we obtained.

First, it is important to point out that we have performed additional experiments to substantiate this part of the work, i.e. a transcriptome analysis with control and cytoPfkfb3 explants cultured in 10 mM glucose. We decided to focus on an early time point, i.e. three-hour after incubation, in order to increase the chance to score the primary response of PSM cells upon changes in glycolytic flux. In addition, our nanostring data in Figure 2—figure supplement 2 shows that glucose titration can change the expression levels of some Wnt-targets in both directions, *i.e.* decreasing glucose upregulates their expressions while increasing glucose downregulates their expressions. Again, this analysis was done at short time-scales to score the immediate effect.

One possible explanation regarding the difference to Oginuma et al. could indeed be the late time point of analysis in their study, i.e. 16-hour after culture. This difference in sampling time, i.e. 3-hour vs. 16-hour after culture, is of particular importance given the dynamic nature of metabolic and signaling responses.

We have added a sentence to explain this point in more detail (line 414-419):

“This discrepancy could relate to the time point of analysis: while Oginuma et al. mainly focused on analyzing samples 16-hour after metabolic changes, we chose to score the effects of altered glycolytic flux/FBP levels already after a three-hour incubation, with the goal to capture the primary response of PSM cells. Whether the difference in sampling time underlies the observed difference is yet unknown, but both studies highlight that Wnt signaling is responsive to glycolytic flux, supporting a tight link between metabolism and PSM development.”

Minor comments.It was not specified the tissue used for the Western blot analyses (was it the PSM alone, the whole tails including somites, etc). This is of relevance to comment #3.

#20. PSM explants without somites were cultured for one/three-hour and were subjected to subcellular protein fractionation. This information is now included in the revised method section.

Reviewer #2 (Significance (Required)):– The work described in this manuscript identifies FBP as a sentinel metabolite for the glycolytic flux. This, itself has the potential to be important for different processes in which differences in glycolysis makes a difference, although I do not think that this will be relevant for the developmental process on which the authors focused their study (see major comments #1 and 2). Indeed, the lethality of global transgenic cytoPfkfb3 expression (although it was not analyzed if it was during development of in postnatal stages, or the cause of this lethality) but with very minor effects on somitogenesis in vivo supports this conclusion.

#21. Please see our detailed comments also based on the newly added in vivo experiments done with the Akita diabetic mouse model in our responses #9–14.

– The potential moonlighting activity of Pfk (connected with specific subcellular localization), is an interesting idea but so far does not go beyond pure speculation. This is prone to the typical double edged effect of stimulating research in that direction but also the potential negative effect of being taken for granted without rigorous proof.

#22: We have added a statement to highlight the nature of this finding and the requirement for follow up studies both in this and other contexts. Please refer to our response #15 for the details.

– The importance of metabolism in general and glycolysis in particular for somitogenesis and axial extension has been recently reported (the relevant papers are cited in the manuscript) and therefore the work described in this manuscript extends those studies. Also, the recent observations that metabolic process can influence cell activity beyond their participation on the classical pathways in which they are involved, including processes apparently as distant as epigenetic regulation of gene activity (see for instance Tarazona and Pourquie, 2020, Dev Cell 54, 282-292), is opening new perspectives to the study of the influence of metabolism on physiological and pathological processes (championed by cancer and immunological response). It also provides a link between control mechanisms across large scale phylogeny, from procaryotes to eukaryotes.– In principle, the potential audience for this work could be wide, as the interest in understanding the involvement of metabolism in the regulation of physiological and pathological processes has been growing over the last years. However, the lack of proven mechanism for the activity of FBP might restrict the real general impact of this work. In this regard, the suggestion that it might control some type of still unknown moonlighting activity of Pfk is so far totally speculative.– I am a developmental biologist with strong focus on mechanisms of somitogenesis and axial extension in vertebrate embryos. There is no part of this work for which I do not feel competent to evaluate.Reviewer #3 (Evidence, reproducibility and clarity (Required)):SummaryIn the present manuscript, Miyazawa and colleagues explore the role of glycolytic flux on embryonic development by using presomitic mesoderm (PSM) patterning as a model.First, the authors examined the steady-state levels of central carbon metabolism metabolites in PSM explants. Explants were cultured in various concentrations of glucose and subjected to gas chromatography mass spectrometry (GC-MS). These experiments allowed the identification of metabolites (such as lactate, 3PG, and FBP) that exhibit a linear correlation with glucose levels and can therefore serve as sentinel metabolites for glycolytic flux in PSM cells. Among the metabolites identified, fructose 1,6-bisphosphate (FBP) showed the strongest linear correlation with glucose levels and was used to inform the design of subsequent experiments.Second, to elucidate the functional role of FBP on PSM patterning, the authors supplement the media used to culture PSM explants with various concentrations of FBP and:- analyze the dynamics of Notch signaling (a critical player in mesoderm segmentation during embryogenesis) using real-time imaging of the LuVeLu reporter;- assess gene expression patterns using in situ hybridization of candidate genes.The authors find that supplementation with FBP, but not F6P or 3PG, impairs mesoderm segmentation and disrupts the activity of the segmentation clock in the posterior PSM. Furthermore, FBP supplementation led to the reduced expression of FGF- and WNT-target genes Dusp4 and Msgn, respectively.Third, the authors generate a conditional cytoPfkfb3 transgenic mouse line in which a cytoplasmic form of the Pfkfb3 enzyme is overexpressed. Pfkfb3 can promote glycolysis, and more importantly, leads to increased levels of FBP in a glucose-dependent manner. The authors find that cytoPfkfb3 transgenic PSM explants contain higher levels of FBP and secrete lactate at higher levels when compared to control explants. Importantly, cytoPfkfb3 transgenic PSM explants exhibit impaired somite formation and reduced expression of Msgn (but not Dusp4) in a glucose-dependent manner when compared to control explants.Finally, the authors investigate changes in protein subcellular localization in their pharmacological and genetic models of FBP-driven glycolytic flux activation. This was prompted by previous reports on the changes in subcellular localization of glycolytic enzymes (Hu et al., 2016). To this end, the authors perform proteome-wide cell-fractionation analyses in drug-treated and cytoPfkfb3 transgenic PSM explants and find that certain glycolytic proteins exhibit altered subcellular localization in both cases (albeit in different fractions).Major concerns:(Re: Results from Figure 2 and Figure S1.)– Given that FBP levels are highly correlated with extracellular glucose levels (which impact glycolytic flux )(TeSlaa and Teitell, 2014) the authors should elaborate on why progressive increase in extracellular glucose does not affect PSM patterning, in the same way that increasing FBP levels does. This is especially important given the claim that FBP is a sentinel metabolite of glycolytic flux.

#23. This important point was also addressed by the reviewer 2, so please see our responses that are also listed under #9, #10, #14 (below).

(Our response #9)

We agree with the reviewer and to directly address this central point, we have performed an extended, additional experiment, collecting 375 embryos to quantify FBP levels under five conditions with three biological replicates.

There are two major results that we highlight here: First, we found that addition of F6P did not lead to increased FBP levels compared to control samples cultured in 10 mM glucose, which is in stark contrast to cytoPfkfb3 embryos cultured in 10 mM glucose (revised Figure 3E). Second, while increasing glucose concentration is mirrored by elevated FBP levels as we reported, we find clear evidence of saturation above a concentration of 10mM glucose: increasing glucose to 25mM does not increase FBP levels further (revised Figure 3E).

This saturation effect seen in glucose titration, but also the absence of elevated FBP upon F6P addition, might be expected outcomes because, as also the reviewer 1 pointed out in the response, Pfk is commonly considered to be a rate-limiting enzyme in the glycolytic pathway. We now have the direct experimental data supporting this hypothesis and thank the reviewers to have initiated this additional (very involved..) experiment.

This new data allows us to conclude more firmly on the correlation between FBP levels and phenotype: at high FBP levels, which are seen in cytoPfkfb3 samples, we observe PSM patterning defects. These high levels are not reached even at 25mM glucose or upon F6P addition, due to the saturation at the level of PFK enzymatic step. Hence, while glucose titration does elevate FBP significantly until this saturation, FBP levels are not as high as in cytoPfkfb3 samples. As a correlative finding, we see that only those conditions with very high FBP levels, or the direct addition of high levels of FBP, cause the arrest of segmentation clock activity. At moderately elevated FBP levels, observed in control explants with high glucose or in cytoPfkfb3 explants with low glucose, clock activity continues and we find a quantitative effect at the level of gene expression, i.e. Wnt signaling target downregulation (Figure 5A, Figure 2—figure supplement 2).

The new data has been included in the revised manuscript and the text has been adjusted accordingly:

– (Result Part, line 178–184) "Consistently, we found that cytoPfkfb3 overexpression lifted the upper limit of FBP levels in PSM cells (Figure 3E, Figure 3—figure supplement 1B, Figure 3—figure supplement 1C). In control explants, FBP levels did not increase further when glucose concentration was increased from 10 mM to 25 mM. It was also the case when control explants were cultured in 20 mM of F6P (Figure 3E). These results indicate that the Pfk reaction carries a (rate-)limiting role for glycolytic flux and FBP levels, and that cytoPfkfb3 overexpression hinders the flux-regulation function of Pfk."

– (Discussion Part, line 378–391) “Our findings suggest that flux-regulation at the level of Pfk is critical to keep FBP steady state levels within a range compatible with proper PSM patterning and segmentation. In agreement with such a rate-limiting function for Pfk, we found in glucose titration experiments that FBP levels saturated and did not further increase at glucose levels above 10 mM (Figure 3E). Along similar lines, the supplementation of high concentrations of the Pfk substrate F6P did not result in a significant increase of FBP levels, again compatible with a rate-limiting function at the level of Pfk (Figure 3E). The upper limit of glycolytic flux and FBP levels can be experimentally increased by cytoPfkfb3 overexpression (Figure 3B, 3E). We interpret the data as evidence that cytoPfkfb3 overexpression compromises the flux-control function of Pfk and hence much higher FBP (and secreted lactate) levels are reached. Such a drastic increase in glycolytic flux and FBP levels correlates with a severe PSM patterning phenotype (Figure 4), which resembles the phenotype induced by supplementation of high dose of FBP (Figure 2). Our results in mouse embryos hence provides evidence that flux regulation by Pfk, an evolutionary conserved role present from bacteria to humans, serves to maintain FBP levels below a critical threshold.”

(Our response #10)

#10. First, we would like to clarify that while indeed glycolytic activity is graded along the PSM, as other and we reported previously (reported in Bulusu et al., 2017 and Oginuma et al., 2017), the baseline expression of the entire glycolytic machinery (from glucose transport to lactate production) is very high, in all PSM cells. Hence, we see that cells all along the entire PSM have very active glycolysis, the posterior PSM being even more active.

For this and related reasons, our interpretation about the difference seen between glucose titration/F6P addition on one side, and FBP addition/cytoPfkfb3 addition on the other side, is based on the role of Pfk in controlling either flux levels or dynamics in all PSM cells.

Hence, while we agree that we generate experimental conditions that allow FBP levels to surpass those found in control embryos, we would like to highlight the fact that even moderate changes in flux does result in very robust functional consequences on gene expression (Figure 2—figure supplement 2, Figure 5), as we show in this work.

We can currently not fully address the first point raised, i.e. the role of graded flux/graded metabolite levels, due to the experimental limitations. Such a study requires, for instance, the generation of metabolite biosensor reporter lines in order to be able to monitor these changes dynamically, in space and time.

(Our response #14)

In our view, two main conclusions, both in vivo and in vitro, can be drawn based on the result we obtained:

First, we find that a moderate increase in glycolytic flux, within the physiological range, leads to a quantitative and consistent change in gene expression, such as downregulation of Wnt target genes (Figure 2—figure supplement 2, Figure 5). Such a phenotype was the result of either glucose titration or culturing cytoPfkfb3-transgenic embryos in low glucose concentration.

In these conditions, while overall PSM patterning is qualitatively normal, we do find consistent changes at quantitative level, i.e. gene expression changes, which are also mirrored by a reduced rate of segmentation (Figure 4B). A detailed analysis of the quantitative changes at the level of segmentation clock dynamics is being carried out and will be presented in a dedicated follow up study.

Second, we find that a very significant increase in FBP levels, i.e. when cytoPfkfb3 transgenic animals are cultured in high glucose conditions or when samples are cultured in high levels of FBP, PSM patterning is qualitatively altered and segmentation clock ceases to oscillate. In this case, we agree that it is not a physiological condition, as such high levels of flux and FBP are not reached in control samples which have intact flux regulation by Pfk. Nevertheless, such an experimental condition can be insightful, as it very clearly reveals the potential link between glycolysis, clock activity, PSM patterning and the Wnt signaling pathway.

It is the combination between the moderate and the more severe effects, observed both in vitro, and now also in vivo using the Akita model (see above), that we take as evidence for an intrinsic, physiological link between glycolytic activity, PSM patterning and signaling.

(Re: Figure 2A and Figure 2B)– The authors should be consistent with the glucose concentrations for the experiments where they assess the dynamics of Notch signaling (Figure 2A) and gene expression (Figure 2B) or otherwise elaborate on why different concentrations are used for these assays.

#24: We agree that ideally the experimental parameters should be as consistent as possible. In regards to the control glucose concentration used in this study, both 0.5 mM and 2.0 mM glucose were used. It reflects that over the years, minor adjustments in the experimental protocol were made, i.e. we now use 2.0 mM glucose as standard setting for all experiments, while previously, 0.5 mM glucose was used (see Bulusu et al., 2017). This change is based on the observation of a slightly improved culture outcome, in terms of reporter gene expression. We have confirmed that the developmental outcome and also effects seen upon addition of FBP are consistent at 0.5 mM and at 2.0 mM glucose. We made a note in the methods section to explain this point (line 513-515):

“Basal culture condition was 0.5 mM glucose at the beginning of this study but was later switched to 2.0 mM glucose which yields a slightly improved reporter gene expression. No major difference was observed in the effects of FBP between these glucose conditions.”

(Re: Results from pharmacological and genetic models of increased FBP levels)– The authors state that FBP-driven impairment of mesoderm segmentation is most pronounced in the undifferentiated PSM cells (in the posterior-most end of the explants) and is, therefore, unlikely to be due to a toxic effect that might otherwise affect the whole explant. While this is a reasonable assumption, it does not discount the possibility that the spatial specificity of the effect of FBP could be driven primarily by increased cell death in the posterior end of the explant. Thus, the authors should test whether cell death underlies the mesoderm patterning defects seen in PSM explants subjected to increased FBP levels.

#25. We have performed immunostaining of active caspase-3 in explants cultured for three-hour in medium containing 0.5 mM glucose and 20 mM FBP and found no difference between control and FBP-treated explants (please refer to Figure 2—figure supplement 1C). This qualitative result does not indicate a major effect via cell death in the tail bud region (i.e. posterior PSM) as the underlying reason for the observed phenotype. We included the new data in the revised Figure 2—figure supplement 1C and adjusted the text accordingly.

(Re: Gene expression experiments/analyses)– This study would benefit greatly from transcriptomic analysis of wt and cytoPfkfb3 transgenic PSM explants (and/or transcriptomic characterization of FBP-treated vs. control PSM explants). The candidate approach used to assess gene expression (through in situ hybridization) may not be sufficient to conclude that cytoPfkfb3 over-expression leads to the downregulation of Wnt signaling (a claim the authors make at the beginning of the manuscript).

#26. We fully agree with the reviewer’s comment. We have now performed RNA-sequencing (RNAseq) analysis using control and cytoPfkfb3 explants cultured in 10 mM glucose, importantly after three hours of incubation in order to score early effects at transcriptome level (please refer to Figure 5C–E).

We found clear evidence that many Wnt-target genes (i.e. *Axin2, Cdx4, Dact1, Dkk1, Mixl1, Msgn1, Sp5, Sp8, T*) were significantly downregulated in cytoPfkfb3 explants, supporting the conclusion that Wnt signaling activity is downregulated in cytoPfkfb3 explants under high glucose condition.

Furthermore, in order to examine similarities between the effects of cytoPfkfb3 overexpression and FBP supplementation, we also performed RNAseq analysis with explants treated with high dose of FBP or F6P. FBP supplementation resulted in downregulation of Wnt target gene expression (i.e. *Dact1, Dkk1, Mixl1, Lef1, Sp5, T, Tbx6*), mirroring the effects seen in cytoPfkfb3 samples. Such a response was not detected in F6P-treated explants.

Combined, these new data significantly strengthen our conclusion that an increase in glycolytic flux and FBP levels leads to downregulation of Wnt signaling activity. The new data is now included in the revised Figure 5C–E and adjusted the texts accordingly.

(Re: Results related to the neural tube closure defects in cytoPfkfb3 transgenic embryos)– The section of the manuscript describing the neural tube closure defects in cytoPfkfb3 transgenic embryos is superficial, lacks detail, and distracts from the focus of the study. Perhaps the data and text on neural tube closure defects should be included as supplemental information.

#27: We agree with the reviewer that in the previous version, this data appeared isolated. It also connects with the point raised by the reviewer 2 about the in vivo significance of our findings. To address both these points, we have now performed additional in vivo experiments using a diabetic mouse model (Akita) to directly test the in vivo consequence of cytoPfkfb3, which interestingly links to the previous findings of neural tube defects. Please see our response #13 for the details (below):

(Our response #13)

First of all, we would like to emphasize that the phenotype seen in cytoPfkfb3 embryos, i.e. the reduction of segmentation and downregulation of Wnt-target gene expression, occurs in a glucose dose dependent manner (Figure 4B and 5A). Hence, it is not the overexpression of cytoPfkfb3 per se that can account for the effects seen. But rather, increased glycolytic flux caused by the combination of transgene expression with high glucose results in functional consequences.

In addition, ‘other possible effects’ that the reviewer is referring to should be evident in all transgenic embryos, irrespective of glucose dose. To the contrary, transgenic embryos cultured in low glucose conditions appear unaltered to control embryos.

Second, we agree that we need to distinguish between strong phenotypes, visible at the level of clock arrest, and milder phenotypes, visible at the level of quantitative gene expression changes. It is important to note that the moderate phenotype, i.e. the quantitative gene expression changes seen in posterior PSM, are seen upon the addition of FBP at moderate levels and upon in glucose titration within the physiological concentration range, as well as in cytoPfkfb3 embryos. We take this as evidence that the effects seen in cytoPfkfb3 transgenic embryos reflect a common response also seen under physiological conditions.

To extend this argument to the in vivo setting, we have performed additional experiments using a genetic mouse model for diabetes. As shown in our previous submission, cytoPfkfb3 transgenic animals do not exhibit a drastic in vivo phenotype when dissected at embryonic day 10.5. One interpretation of this finding is that since the cytoPfkfb3 phenotype is glucose and flux-dependent, the in vivo flux is low, reflecting low glucose concentrations described in vivo. To test the effect of increased flux in cytoPfkfb3 embryos in vivo, we therefore crossed the transgenic mice into a diabetic model called Akita, in which a point mutation in the Insulin2 gene causes high maternal glucose levels (Yoshioka et al., 1997; Wang et al., 1999). Using this experimental setup, we tested whether transgenic embryos in Akita diabetic females would manifest in vivo phenotypes.

Indeed, we found that cytoPfkfb3 transgenic embryos developing in Akita diabetic females showed significantly increased cases of neural tube closure defects (50% of cytoPfkfb3 embryos) and developmental delay (control: 38 somites vs. cytoPfkfb3: 34 somites at E10.5), defects not seen in transgenic cytoPfkfb3 embryos from control females (please refer to Figure S4—figure supplement 1D-E ). This dependency of the in vivo phenotype on maternal glucose conditions again highlights that the defects observed in cytoPfkfb3 embryos are not due to the expression of cytoPfkfb3 per se, but are rather directly linked to increased/unregulated glycolytic flux.

We included the new in vivo data in the revised Figure S4—figure supplement 1D-E and modified the text accordingly*.*

(Re: Conclusions of the study)– A previous study by Oginuma et al., 2020 provided strong evidence for a mechanism underlying the positive regulation of Wnt signaling by glycolysis (initiated by the elevation of intracellular pH) in the chick embryo tailbud. As mentioned in the discussion, the results of the present study are not consistent with this mode – and this contradiction is not sufficiently resolved. This is a concern, given that the evidence that cytoPfkfb3 inhibits Wnt signaling is sparse (see above).

#28: This important point was also raised by the reviewer 2, please see our response as listed under #19 (below).

(Our response #19)

We cannot resolve this discrepancy, but now offer a more detailed discussion, also based on the additional data we obtained.

First, it is important to point out that we have performed additional experiments to substantiate this part of the work, i.e. a transcriptome analysis with control and cytoPfkfb3 explants cultured in 10 mM glucose. We decided to focus on an early time point, i.e. three-hour after incubation, in order to increase the chance to score the primary response of PSM cells upon changes in glycolytic flux. In addition, our nanostring data in Figure 2—figure supplement 2 shows that glucose titration can change the expression levels of some Wnt-targets in both directions, i.e. decreasing glucose upregulates their expressions while increasing glucose downregulates their expressions. Again, this analysis was done at short time-scales to score the immediate effect.

One possible explanation regarding the difference to Oginuma et al. could indeed be the late time point of analysis in their study, i.e. 16-hour after culture. This difference in sampling time, i.e. 3-hour vs. 16-hour after culture, is of particular importance given the dynamic nature of metabolic and signaling responses.

We have added a sentence to explain this point in more detail (line 414-419):

“This discrepancy could relate to the time point of analysis: while Oginuma et al. mainly focused on analyzing samples 16-hour after metabolic changes, we chose to score the effects of altered glycolytic flux/FBP levels already after a three-hour incubation, with the goal to capture the primary response of PSM cells. Whether the difference in sampling time underlies the observed difference is yet unknown, but both studies highlight that Wnt signaling is responsive to glycolytic flux, supporting a tight link between metabolism and PSM development.”

– Another discrepancy lies in the lack of an observable phenotype when culturing mouse PSM explants at very low glucose concentrations (e.g., 0.5 mM in Figure 2A). Oginuma et al. observed clear disruptions to embryonic elongation and somite formation at a glucose concentration equal to 0.83 mM. Would this be due to species-specific mechanisms? Furthermore, while the authors focus on sentinel metabolites (such as FBP), experiments involving direct manipulation in glycolysis could resolve some of these inconsistencies.

#29: Indeed species specific differences in the requirement for glucose are to be expected. Our extensive analysis shows that at 0.5mM glucose, segmentation and elongation proceeds (Bulusu et al., 2017).

Regarding the second point, we have outlined several strategies to directly perturb glycolysis, i.e. glucose titration (mirrored by increase in lactate secretion) and by genetic targeting of the rate-limiting enzyme, Pfk. Glucose titration in wild-type embryos corresponds to the experiment the reviewer suggested, and we again found that higher glucose (i.e. higher flux) leads to down regulation of several Wnt-target genes (Figure 2—figure supplement 2). Of note, also in cytoPfkfb3 explants the effects are glucose-dose dependent (again mirrored by increase of lactate secretion), clearly indicating that we successfully and directly controlled glycolysis.

References1. Hu, Hai, et al. "Phosphoinositide 3-kinase regulates glycolysis through mobilization of aldolase from the actin cytoskeleton." Cell 164.3 (2016): 433-446.2. TeSlaa, Tara, and Michael A. Teitell. "Techniques to monitor glycolysis." Methods in enzymology 542 (2014): 91-114.3. Oginuma, Masayuki, et al. "Intracellular pH controls WNT downstream of glycolysis in amniote embryos." Nature584.7819 (2020): 98-101.Reviewer #3 (Significance (Required)):The experimental results reported in this study enhance our understanding of how cellular metabolic states regulate cellular behaviors during embryonic development. The study provides insight into how PSM elongation is controlled by morphogenetic mechanisms that are modulated by glycolytic flux. One of the strengths of the study is the use of an interdisciplinary approach that includes GC-MS, in vivo imaging and mouse transgenic lines. It should be noted that some of the conclusions of the study diverge from previous papers that examine the role of metabolism in developmental patterning (e.g., Oginuma et al., 2020).